



# Superposition of gravity waves with different propagation characteristics observed by airborne and space-borne infrared sounders

Isabell Krisch[1,*], Manfred Ern[1], Lars Hoffmann[2], Peter Preusse[1], Cornelia Strube[1], Jörn Ungermann[1,3], Wolfgang Woiwode[4], and Martin Riese[1]

[1]Forschungszentrum Jülich, Institute of Energy- and Climate Research, Stratosphere (IEK-7), Jülich, Germany
[2]Forschungszentrum Jülich, Jülich Supercomputing Centre, Jülich, Germany
[3]Jülich Aachen Research Alliance (JARA), Jülich, Germany
[4]Karlsruhe Institute of Technology, Institute of Meteorology and Climate Research - Atmospheric Trace Gases and Remote Sensing (IMK-ASF), Karlsruhe, Germany
[*]now at Deutsches Zentrum für Luft- und Raumfahrt, Institut für Physik der Atmosphäre, Oberpfaffenhofen, Germany

**Correspondence:** Isabell Krisch (isabell.krisch@dlr.de)

**Abstract.** A complex gravity wave structure consisting of a superposition of multiple wave packets was observed above southern Scandinavia on 28 January 2016 with the Gimballed Limb Observer for Radiance Imaging of the Atmosphere (GLORIA). The tomographic measurement capability of GLORIA enabled a detailed 3-D reconstruction of the gravity wave field and the identification of multiple wave packets with different horizontal and vertical scales. The larger-scale gravity waves with hori-

zontal wavelengths around 400 km could be characterised using a 3-D wave-decomposition method. For the characterization of the smaller-scale wave components with horizontal wavelengths below 200 km, the 3-D wave-decomposition method needs to be further improved in the future.

For the larger-scale gravity wave components, a combination of gravity-wave ray-tracing calculations and ERA5 reanalysis fields identified orography as well as a jet-exit region and a low pressure system as possible sources. All gravity waves propagate

upward into the middle stratosphere, but only the orographic waves stay directly above their source. The comparison with ERA5 also shows that ray-tracing provides reasonable results even for such complex cases with multiple overlapping wave packets. AIRS measurements in the middle stratosphere support these findings, even though their coarse vertical resolution barely resolves the observed wave structure in this case study. The high-resolution GLORIA observations are therefore an important source of information on gravity wave characteristics in the upper troposphere and lower stratosphere region.

## 1 Introduction

Gravity waves (GWs) are an important coupling mechanism in the atmosphere as they can transport energy and momentum over large horizontal and vertical distances. Even though they were discovered in the first half of the 20th century (Wegener, 1906; Trey, 1919), many processes regarding their sources, propagation and dissipation are still not fully understood (Alexander et al., 2010; Geller et al., 2013; Plougonven and Zhang, 2014). Due to this lack of understanding and because of computational





constraints, gravity waves are over-simplified in current numerical weather prediction and climate projection models by employing parametrisation schemes. This leads to large uncertainties in the surface temperature, surface pressure and middle atmosphere circulation characteristics (Sigmond and Scinocca, 2010; McLandress et al., 2012; Shepherd, 2014; Sandu et al., 2016; Garcia et al., 2017).

To improve our understanding of gravity wave processes and especially their propagation characteristics, measurements are

required that allow for a full wave characterization and make wave propagation studies possible. So far several measurement techniques have been developed to fully characterise gravity waves. For example, in-situ measurements of close-to-vertical profiles can be analysed using hodograph analysis, the Stokes method, or a combination of wind and temperature measurements to fully characterise gravity waves (Eckermann and Vincent, 1989; Guest et al., 2000; Wang and Geller, 2003; Zhang et al., 2014). Furthermore, techniques based on horizontal 1-D measurements, e.g. from airplanes or super pressure balloons, have

also been used to derive gravity wave characteristics (Hertzog et al., 2008; Fritts et al., 2016; Gisinger et al., 2020). All these methods rely on the polarisation and dispersion relation to infer the wave structure and usually do not show the 3-D distribution and spatial change of wave characteristics.

Recently, new remote sensing techniques have been employed to obtain the 3-D structure of gravity waves directly using spaceborne or airborne temperature measurements (Ern et al., 2017; Krisch et al., 2017; Wright et al., 2017). One of these

new measurement techniques is 3-D tomography with the Gimballed Limb Observer for Radiance Imaging of the Atmosphere (GLORIA; Riese et al., 2014; Friedl-Vallon et al., 2014). GLORIA is a limb-viewing infra-red spectrometer that can scan the atmosphere by panning its horizontal viewing direction. By combining multiple measurements under different viewing angles, this technique is capable of nicely reproducing the 3-D structure of mesoscale gravity waves in the upper troposphere / lower stratosphere (UT/LS) region (Krisch et al., 2017, 2018; Krasauskas et al., 2019). The data acquisition method using GLORIA,

as well as the data processing are explained in detail in Section 2.

So far, GLORIA has only been used to investigate gravity waves with one dominant wave component. However, in-situ measurements showed in many cases the presence of a large spectrum of gravity waves within the same measurement volume (e.g. Smith et al., 2016; Smith and Kruse, 2017; Portele et al., 2018). In Section 3 it will be examined, if tomographic GLORIA measurements are also capable of reproducing complex wave patterns with a superposition of multiple wave packets in the

same measurement volume.

3-D spectral analysis is required to determine the gravity wave characteristics from 3-D temperature measurements. Commonly used techniques are either a 3-D S-Transform (Wright et al., 2017) or a 3-D wave fitting algorithm called S3D (Lehmann et al., 2012). In this paper, the S3D method will be used to differentiate between multiple wave packets within the same measurement volume. Additionally, it will be investigated if such wave characterisation results can be used to determine the various

sources of these wave packets (Sec. 4).

The propagation paths of the gravity waves will be identified using the Gravity wave Regional Or Global RAy Tracer (GROGRAT Marks and Eckermann, 1995; Eckermann and Marks, 1997). Ray-tracing methods are typically based on linearisation and are usually only valid if one wave packet propagates through a background field with variations that are large compared to the wavelengths. This paper will study if such simplified methods also produce reasonable results in more complex cases with



multiple wave packets by comparing ray-tracing results with meteorological reanalysis data and satellite measurements in the
mid-stratosphere (Sec. 4).

## 2 Methodology and data description

### 2.1 Tomographic measurement concept of GLORIA

The gimballed limb observer for radiance imaging of the atmosphere (GLORIA) is an airborne instrument, which measures
the infrared radiation emitted by atmospheric trace species and particles (Friedl-Vallon et al., 2014; Riese et al., 2014). This is
accomplished by combining a 2-D detector array with a Michelson Interferometer. In this way, GLORIA can measure 48x128
infrared spectra simultaneously every two seconds. These spectra cover the spectral range between $780\,\mathrm{cm}^{-1}$ and $1400\,\mathrm{cm}^{-1}$
($7\,\mu m$ to $13\,\mu m$), thus allowing the measurement of emissions by a multitude of atmospheric trace species. As clouds are
usually opaque in the spectral range of GLORIA, trace gas measurements can only be taken in sufficiently cloud free layers of
the atmosphere.

GLORIA looks to the right with respect to the flight direction. A linear flight path therefore provides 2-D curtains of
temperature and trace gases. Furthermore, GLORIA has the unique ability to pan its line-of-sight (LOS) between $45°$ and
$135°$ with respect to the aircraft heading, which enables a horizontal scanning of the atmosphere. In this mode, GLORIA can
measure the same air volume under different angles. These measurements can be combined using tomographic methods to
reconstruct 3-D fields of the atmospheric temperature and 3-D trace gas distributions (Ungermann et al., 2011; Krisch et al.,
2018). GLORIA's tomographic measurement concepts can be divided into two groups: full angle tomography (FAT) and limited
angle tomography (LAT). In FAT, the investigated volume is measured from all sides using closed flight patterns, e.g. circles. In
contrast, LAT uses measurements from only a limited set of angles and can be applied already on linear flights or half circles.

FAT can reconstruct cylindrical atmospheric volumes with very high spatial resolutions of up to $20\,\mathrm{km}$ in all horizontal
directions and $200\,\mathrm{m}$ in the vertical (Krisch et al., 2017). However, to fly those circular patterns with sufficient diameter
($\approx 400\,\mathrm{km}$) takes around two hours. Thus, a sufficiently stationary behaviour of the atmospheric flow is required. This poses
some limitations for the observation of GWs that vary quickly in time.

The maximum volume that can be reconstructed with LAT is given by the tangent point distribution (see Fig. 1). Tangent
points of forward or backward looking measurements are closer to the flight path than those with an azimuth angle of $90°$. At
higher altitudes, the tangent points are closer together and thus the horizontal resolution perpendicular flight track is higher.
At the same time the horizontal extent of the area covered by tangent points is smaller at higher altitudes. In the vertical, the
volume covered by tangent points has a banana-like shape with increasing distance to the flight path and increasing horizontal
extent with decreasing altitude. At an altitude of $3\,\mathrm{km}$ below the aircraft, the horizontal extent of the measurement volume
perpendicular to the flight track is on the order of $150\,\mathrm{km}$.

Using LAT, all overlapping measurements of an air parcel are taken less than $15\,\mathrm{min}$ apart which makes this technique
suitable to more dynamic conditions. Thus, LAT is suitable for measurements of transient GWs and GWs in a fast-changing
background wind, whereas FAT will yield high quality reconstructions for steady GWs with close to zero ground-based phase



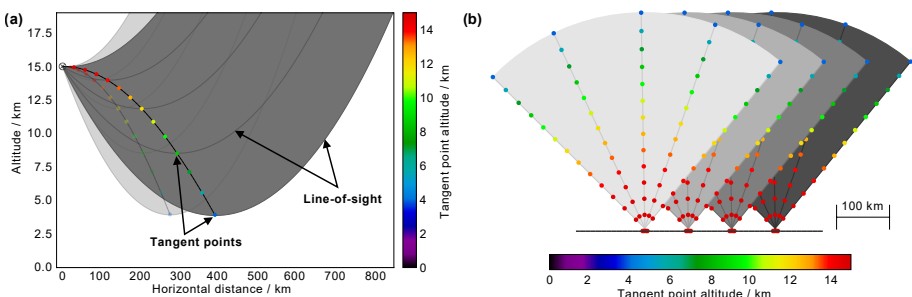

**Figure 1.** (a) Vertical cross-section of the limb-sounding geometry of GLORIA. The flight direction points into into the paper plane. Images taken under $90°$ azimuth cover the dark grey area with the LOS. The respective tangent points (bright coloured dots) increase in distance with decreasing altitude. The tangent points of forward- and rearward-looking images (light grey and pale coloured dots) are closer to the flightpath. The line-of-sight (LOS), which is a straight line in reality, has a parabolic shape in this plot due to the transformation into a Cartesian coordinate system with the x-axis following the Earth surface. (b) Top view in bird perspective of the flightpath of LAT. The dots again indicate the tangent points and are coloured according to their altitude. Each grey sector indicates one horizontal scan from 45 to $135°$. The lighter the grey, the later in time the measurements are taken. Figure taken from (Krisch et al., 2018).

speed. Furthermore, the resolution of LAT is slightly degraded compared to FAT and is only 30 km along the flight track, 70 km perpendicular to the flight track and 400 m in the vertical. A detailed discussion of the advantages and disadvantages of both

methods especially with regards to gravity wave measurements can be found in (Krisch et al., 2018). For the present paper, LAT is applied because the observed gravity wave structure is varying with time.

## 2.2 Temperature retrieval for the Atmospheric Infrared Sounder (AIRS)

The Atmospheric Infrared Sounder (AIRS; Aumann et al., 2003; Chahine et al., 2006) is a nadir-scanning instrument onboard NASA's Earth Observing System (EOS) Aqua satellite that performs scans across the satellite track. Each scan consists of 90

footprints across track, and the width of the swath is about 1800 km. At nadir, the footprint diameter is 13.5 km, and the across-track sampling step is 13 km. The along-track sampling distance is 18 km. The EOS Aqua satellite is in a sun-synchronous orbit with fixed equator crossing times of 13:30 LT for the ascending orbit (flying northward) and 01:30 LT for the descending orbit (flying southward).

AIRS is a hyperspectral sounder that measures atmospheric emissions of $CO_2$ and other trace gases with high spectral

resolution. In contrast to the limb geometry, nadir sounding depends on the optical depth along the line-of-sight to gain vertical information. Depending on the wavelength, the sensitivity function along line-of-sight peaks at different altitudes (Hoffmann and Alexander, 2009). By combining multiple spectral channels, a temperature altitude profile can be retrieved. In contrast to limb sounders, the vertical resolutions of these nadir profiles are usually on the order of 10 km in the stratosphere.

For retrievals of night time data, emissions in the 4.3 μm and the 15 μm spectral bands can be combined. For day time

retrievals only the 15 μm band is used due to non-local thermodynamic equilibrium effects which influence the 4.3 μm band. Correspondingly, AIRS night time data have a better vertical resolution and lower noise. Except at polar latitudes, day time





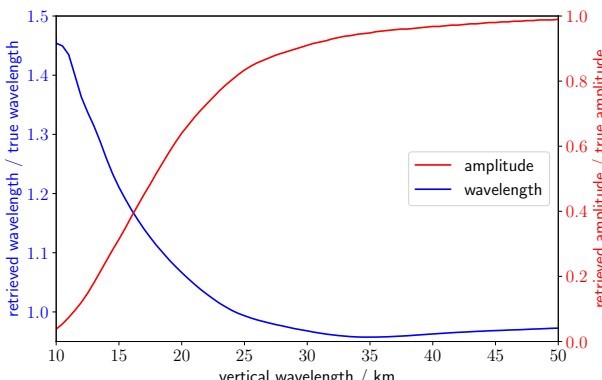

**Figure 2.** Sensitivity function of the AIRS temperature retrieval for GW amplitude (red) and vertical wavelength (blue) as estimated from a 1-D sinusoidal fitting routine at given real vertical wavelengths.

data correspond to ascending orbits, and night time data to descending orbits, respectively. The AIRS temperature retrievals presented in this paper follow the retrieval set-up presented by Hoffmann and Alexander (2009).

The vertical resolution of these temperature retrievals varies from 6.6 km to 14.7 km depending on altitude. The total ac-
curacy lies between 0.6 K and 2.1 K, while the precision is in the 1.5 K–2.1 K range (Hoffmann and Alexander, 2009). The retrieval has been designed for stratospheric altitudes and provides its best results between 20 km and 50-60 km. Validation of the AIRS temperature retrievals was discussed by Meyer and Hoffmann (2014).

In order to allow quantitative assessments of GW parameters derived from measurements, the sensitivity function of the observation technique with respect to GWs with different spatial scales has to be considered (Alexander, 1998; Preusse et al.,
2000; Ern et al., 2005; Alexander et al., 2010; Trinh et al., 2016). It maps the true GW amplitude or momentum flux onto the amplitude or momentum flux observed by the given measurement technique. The AIRS sensitivity function for the used retrieval in the middle stratosphere (36 km) is shown in Fig. 2. For vertical wavelengths below 25 km the temperature amplitude is underestimated. At the same time, the waves are stretched in the vertical by up to 50%. As such, the GW spectrum is shifted towards higher vertical wavelengths and AIRS GW observations of waves with vertical wavelengths below 30 km have to be
treated carefully.

These values do not include effects caused by the scale separation of the measured temperature into background temperature and GW perturbations. Sensitivity functions including the effect of scale separation by an across-track 4th-order polynomial (a standard procedure for nadir sounders) are given, for example, by Meyer et al. (2018) or the supporting information of Ern et al. (2017). Moreover, GWs with horizontal wavelengths of less than 100 km, which may be affected by the limited AIRS
footprint size, are not described by the sensitivity function in Fig. 2.



## 2.3 Analysis and reanalysis model data

Modern numerical weather prediction (NWP) relies on two fundamental components: first, a high-resolution global circulation model (GCM) which includes all processes relevant for weather forecasting and, second, the assimilation of a multitude of different types of measurements. The European Centre for Medium-Range Weather Forecasts (ECMWF) integrated forecast
system (IFS) assimilates measurement data by the 4-D var method. The model is constrained by measurements clustered in 12 hour windows from 09 UTC to 21 UTC and from 21 UTC to 09 UTC the next morning. However, as ECMWF tries to provide timely forecasts, measurement data arriving after 15 UTC or 03 UTC cannot be used for the 12 UTC or 00 UTC runs, respectively. Measurements up to an altitude of $\approx 40\,\mathrm{km}$ are used in the assimilation. ECMWF analysis fields are available every 6 hours. These model fields provide a close to reality background for propagation and also trigger realistic excitation of
gravity waves by processes resolved by the model, i.e. mesoscale orography and spontaneous adjustment. Other gravity wave source processes such as convection are parametrized in the GCM and the emitted gravity waves are less realistic (Preusse et al., 2014). It has to be noted, that the assimilation does not constrain gravity waves themselves, thus, they can develop freely from the model physics.

The dynamical core of the ECMWF GCM is based on a spectral representation of the atmosphere. The spatial resolution was
enhanced several times in the recent decade. The ECMWF analysis from 2016 used in this paper uses 1279 spectral coefficients in the horizontal (corresponds to a resolution of $16\,\mathrm{km}$) on 137 levels from the surface up to $80\,\mathrm{km}$. Though the dynamical core would in principal allow to resolve waves with horizontal wavelength double the horizontal resolution, hyperdiffusion, which was introduced to provide numerical stability, limits well-resolved waves to about 10 spatial grid points (Skamarock, 2004; Preusse et al., 2014). Thus, waves of horizontal wavelengths longer than $\approx 150\,\mathrm{km}$ are fully resolved in the ECMWF analysis
fields. Shorter waves, if excited e.g. by topography, may still be present but are suppressed in amplitude.

Besides the above described ECMWF analysis fields, this paper also makes use of ECMWF Reanalysis 5th Generation (ERA5) data. In contrast to the ECMWF analysis runs, ERA5 uses all available measurement data in the 12 hour assimilation windows. Additionally, ERA5 data is available every hour. However, ERA5 has a horizontal resolution of only $31\,\mathrm{km}$ (639 spectral coefficients), which means only gravity waves with horizontal wavelengths larger than $\approx 300\,\mathrm{km}$ are fully resolved.
In summary, the ERA5 reanalysis has a higher temporal, but lower horizontal resolution than the ECMWF operational analysis. Hence, for small scale waves the ECMWF operational analysis is more accurate, but for fast changing situations, ERA5 might be preferable.

## 2.4 Scale separation of atmospheric variables

The atmospheric temperature structure in the mid-latitude stratosphere and troposphere is shaped by dynamical features of
different spatial and temporal scales. The most important features are the mean atmospheric temperature, global and synoptic scale planetary waves and small-scale processes including GWs. The mean atmospheric temperature is governed by slow radiative processes and large-scale meridional circulations. These vary slowly in altitude and latitude, but are assumed to remain constant in zonal direction. Planetary waves surround the Earth on latitude circles. Thus, they have integer zonal wave





numbers. In the mid stratosphere, the main planetary wave modes have zonal wave numbers of 1-6. In the lower stratosphere

and troposphere, also planetary waves with higher zonal wave numbers exist. GWs have horizontal wavelength scales of a few

kilometres to several thousand kilometres. However, due to the resolution of GLORIA measurements and the spatial extent

of the measurement volume, we will focus here on the identification of mesoscale GWs with horizontal wavelengths between

$\approx 100\,\mathrm{km}$ and $\approx 1000\,\mathrm{km}$.

For global data sets, background and GW fluctuations are often separated using zonal filtering with a cut-off wave number

of 6 in the mid-stratosphere (e.g. Fetzer and Gille, 1994; Ern et al., 2006, 2018). As the region of interest in this paper is given

by the GLORIA measurement altitude, which is in the lower stratosphere and upper troposphere, zonal filtering with a higher

cut-off wave number 18 is required (Strube et al., 2020) and used for all global datasets (ECMWF and ERA5). As this zonal

filter still might allocate GW structures with long zonal but short vertical and/or meridional wavelengths to the background, a

sliding polynomial smoothing with a Savitzky-Golay filter (SG-filter; Savitzky and Golay, 1964) in the vertical and meridional

direction is applied additionally to the background field to suppress these small scale signals: for the analysis and reanalysis

model data used in this paper, a 4th order SG-filter over a window of $5\,\mathrm{km}$ in the vertical direction and a 3rd order SG-filter

over a window of $750\,\mathrm{km}$ in the meridional direction are used. By subtracting the smooth background temperature from the

total temperature, one receives a perturbation field containing different small scale processes like GWs or different weather

systems like convection or fronts.

Due to the local nature of GLORIA measurements, global filtering algorithms, like the zonal method described above, are not

suitable. Different local filtering methods for GLORIA-like data sets were tested (App. A) and best results were achieved with

three sequentially applied 3rd order SG-filters with windows of $750\,\mathrm{km}$ in each horizontal and $3\,\mathrm{km}$ in the vertical direction.

## 2.5   Spectral analysis using a three-dimensional sinusoidal fitting routine (S3D)

To characterise the temperature perturbations obtained from the scale separation described in the previous section with regard

to GWs, wave parameters (horizontal and vertical wavelengths, wave amplitude and wave direction) are derived. For this task,

a small-wave decomposition method called S3D was used (Lehmann et al., 2012). S3D uses a least square approach to fit a

sine function to the 3-D temperature perturbation field $T'(\boldsymbol{x})$:

$$\chi^2 = \sum_i \frac{\left(f\left(\boldsymbol{x}_i\right) - T'\left(\boldsymbol{x}_i\right)\right)^2}{\sigma_f^2\left(\boldsymbol{x}_i\right)} \tag{1}$$

with weighting function $\sigma_f^2\left(\boldsymbol{x}\right)$ and the sine function

$$f(\boldsymbol{x}) = \hat{T} \cdot \sin(\boldsymbol{k}\boldsymbol{x} + \phi) = A \cdot \sin(\boldsymbol{k}\boldsymbol{x}) + B \cdot \cos(\boldsymbol{k}\boldsymbol{x}), \tag{2}$$

with 3-D wave vector $\boldsymbol{k} = (k, l, m)$, temperature amplitude $\hat{T}$, wave phase $\phi$, sine amplitude $A = \hat{T}\cos\phi$, and cosine amplitude

$B = \hat{T}\sin\phi$. To reduce the impact of measurement data with low confidence values, a weighting function $\sigma_f^2$ is used for the

GLORIA data, which is chosen to be 1 if a tangent point exists in the corresponding grid cell of the retrieval and $10^5$ if not.

The method is applied on analysis cubes – small three-dimensional sub-regions of the perturbation field. In each cube, a

superposition of monochromatic sine waves is assumed and determined by fitting. The quality of the fits depends on the cube





size. If the cubes are too large compared to the resulting wavelengths, small fluctuations get masked by larger scale waves. Additionally, the cube should not be too large since real GWs are highly variable and complex, and an approximation with monochromatic waves is only valid inside small areas (Appendix of Krisch et al., 2017). However, if the cubes are too small, the amount of data points is insufficient to uniquely identify the dominant wave structure. Systematic tests with synthetic waves have shown, that cubes covering only 40% of one wave cycle per direction still lead to reasonable results for the wave vector $k$.

The temperature perturbations derived from GLORIA measurements are highly variable in amplitude. To recover these variations and still keep the cube sizes large enough for reasonable fits of the wave vector $k$, a step wise fitting routine is used. First, the wave vector is fitted in large cube sizes and, second, the wave amplitude $\hat{T}$ and phase $\phi$ are determined in smaller cube sizes using the wave vectors from the larger cubes.

## 2.6 Ray-tracing of gravity waves

The Gravity wave Regional Or Global RAy Tracer (GROGRAT; Marks and Eckermann, 1995; Eckermann and Marks, 1997) is used to study the propagation of the observed GWs. GROGRAT was the first GW ray-tracer to implement the full dispersion relation

$$\omega^2 = \frac{(k^2 + l^2)N^2 + f^2 \left(m^2 + \frac{1}{4H^2}\right)}{k^2 + l^2 + m^2 + \frac{1}{4H^2}}. \tag{3}$$

Thus, GWs of all frequencies, including non-hydrostatic GWs as well as GWs with frequencies close to the Coriolis frequency $f$, can be propagated through a spatially slowly varying background atmosphere (Marks and Eckermann, 1995). In a second version of GROGRAT (Eckermann and Marks, 1997), a not only spatially but also temporally varying background atmosphere has been implemented.

The differential equations $\frac{dx_i}{dt} = \frac{\partial\omega}{\partial k_i}$ and $\frac{dk_i}{dt} = \frac{\partial\omega}{\partial x_i}$, $i = 1, 2, 3$, are solved for multiple time steps using Runge-Kutta methods. For each time step, the wave action conservation law and the full dispersion relation are applied to calculated changes in the wave amplitude. Changes of the ground-based frequency due to temporal variation of the background field are implicitly taken into account by this method. Wave dissipation and damping ($\frac{\partial}{\partial t}A \neq 0$) are accounted for in GROGRAT by including turbulent (Pitteway and Hines, 1963) and radiative (Zhu, 1994) damping schemes and saturation (Fritts and Rastogi, 1985).

The spatially and temporally varying background atmosphere has been constructed from 6-hourly ECMWF analysis fields as described in Sec. 2.4. In addition, GROGRAT applies a 3rd order spline interpolation in both space and time. The start parameters necessary to launch GWs into these background fields are obtained by the sinusoidal-fits described in Sec. 2.5.

## 3 Data acquisition and measurement results

### 3.1 Aircraft campaign

From December 2015 to March 2016 an extensive aircraft measurement campaign took place with ground bases in Oberpfaffenhofen, Germany, and Kiruna, Sweden. This campaign was a conglomerate of several campaigns with different scientific





goals, among them to study the full life cycle of GWs (GW-LCYCLE) and to demonstrate the use of infrared limb imaging for GW wave studies (GWEX). The carrier used for this campaign was the German High Altitude and Long Range Research Aircraft (HALO; DLR 2018). This plane is based on the business jet Gulfstream G550 with modifications that allow mounting

a wide variety of scientific equipment.

The scientific payload of HALO during the winter 2015/2016 campaign encompassed three remote sensing instruments: GLORIA in the belly-pod, an upward looking water vapor, cloud and ozone lidar (WALES), and a differential optical absorption spectrometer. In addition, the Basic HALO Measurement and Sensor System (BAHAMAS; Giez, 2012) measuring temperature, pressure and winds at high precision and high temporal resolution as well as a number of in-situ instruments

measuring trace gases were part of the payload. A more detailed overview of all instruments is given in (Oelhaf et al., 2019).

During the campaign, 18 scientific research flights adding up to 156 flight hours were performed covering 20°N to 90°N and 80°W to 30°E. Seven of these scientific research flights contained measurements of GWs. This paper presents and analyses GLORIA measurement results from a gravity wave flight on 28 January 2016 above southern Scandinavia.

## 3.2 Synoptic situation

For the 28 January 2016, the ECMWF-IFS predicted gravity waves above southern Scandinavia. One prominent source of gravity waves in this region are the Scandinavian Mountains also known as Scandes. The Scandes is a mountain ridge running north-south along the complete west coast of Scandinavia. In the southern part close to the flight track, the mean width of the ridge is around $250\,\mathrm{km}$ and the mean elevation is on the order of $1300\,\mathrm{m}$. Due to the ridge's width, the maximum horizontal wavelength of gravity waves generated by this orography should be on the order of $400\text{-}500\,\mathrm{km}$. According to linear wave

theory (e.g. Nappo, 2012), the wind at the surface to generate mountain waves with a horizontal wavelength of $400\,\mathrm{km}$ at a latitude of $60°\mathrm{N}$ has to be at least $8\,\mathrm{ms}^{-1}$. However, waves generated with such slow wind speeds would have very low vertical group velocities and small saturation amplitudes. Both in the forecast of ECMWF-IFS (not shown) and ERA5 reanalysis (Fig. 3 a) the flow over the southern part of the Scandes is around $17.5\,\mathrm{ms}^{-1}$ in the morning of 28 January 2016. According to theory, a gravity wave with a horizontal wavelength of $400\,\mathrm{km}$, which is generated by a flow over orography with such a wind

speed, has a vertical group velocity of $0.86\,\mathrm{kmh}^{-1}$ and needs $14\,\mathrm{h}$ to propagate to an altitude of $12\,\mathrm{km}$ (GLORIA measurement altitude). Thus, the flight time between $17{:}30\,\mathrm{UTC}$ and $22{:}00\,\mathrm{UTC}$, fits very well to this situation. As the orography of the Scandes is composed of mountain ridges with many different heights and widths, a complex wave structure with many different horizontal wavelengths is expected.

Furthermore, a low-pressure system evolved over southern Scandinavia in the morning of the measurement day, which then

moved slowly eastward (Fig. 3 a & b). This low pressure system forced the eastward jet stream in the upper troposphere to slow down and diverge. Thus, a jet exit region was created over the North Sea between Scandinavia and Great Britain (Fig. 3 c). This jet exit region was following the low pressure system slowly eastwards. Both jet-exit regions as well as convective storms, which often accompany low pressure systems, are prominent sources of gravity waves and were located in the vicinity of southern Scandinavia on this day (Fig. 3 d). Hence, the observed gravity waves could be expected to be a mixture of waves

generated by orography, the jet-exit region and convection.

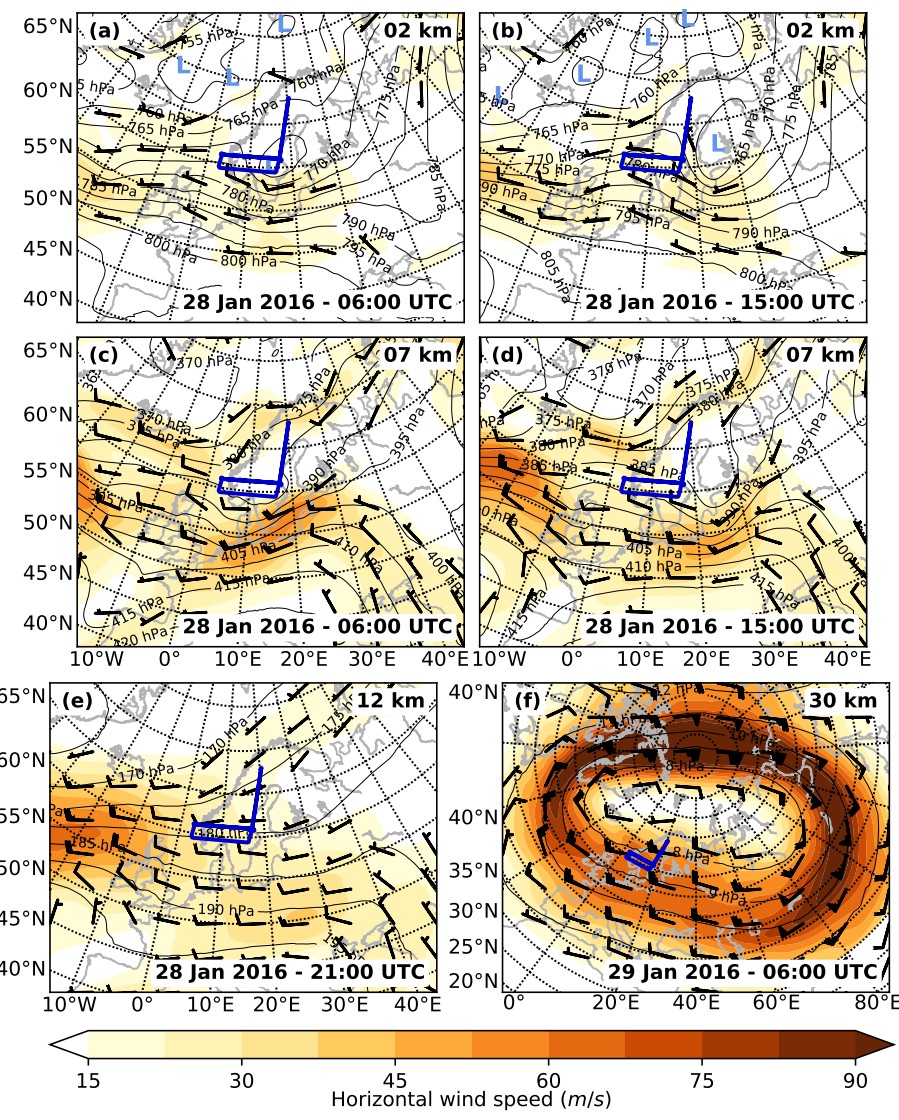

**Figure 3.** Synoptic situation over Northern Europe on 28/29 January 2016. Shown are ERA5 horizontal wind (colour and barbs) and pressure (contour lines) fields at different altitudes and time steps. Low pressure systems are marked with a light blue "L". The altitude of the respective cross section is always given on the top right of the panel, the model time at the bottom right. The dark blue line marks the flight path.

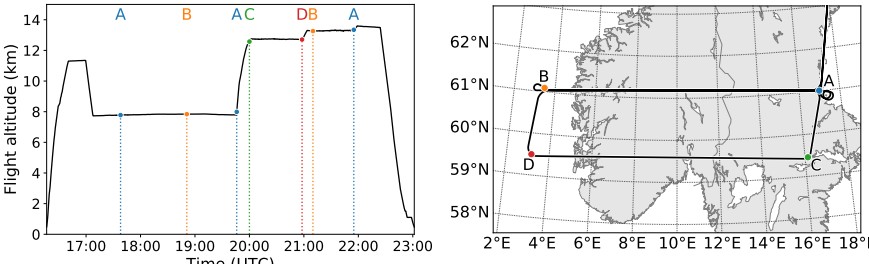

**Figure 4.** HALO flight over Southern Scandinavia on 28 January 2016. The left panel shows the different flight altitudes, the right panel the geographic location. The letters and dotted lines in the left panel mark at which point in time the respective geographic locations in the right panel are reached.

The divergence in the jet stream was also connected with a low tropopause altitude and accordingly a low cloud top height of around 8 km above southern Scandinavia, which results in good measuring conditions for GLORIA. However, it also sharpened the tropopause, which can lead to partial reflection of gravity waves. The horizontal wind kept its eastward orientation at higher altitudes (Fig. 3 e & f) as the maximum of the circumpolar jet stream on this side of the pole was located just south of 
Scandinavia. This provided favourable conditions for vertical GW propagation.

### 3.3   GLORIA measurements and diagnostics

The GW structure was probed with multiple, 700 km long, linear flight legs crossing southern Scandinavia in zonal direction (cf. Fig. 4). To study the interaction of the GWs with the tropopause by in situ observations (Gisinger et al., 2020), two flight legs were positioned below (leg 1 and 2) and two flight legs above the tropopause (leg 3 and 4). Both lower legs were 
performed at 61°N (leg 1 from point A to point B and leg 2 from point B to point A) and were mainly dedicated to in situ and water vapour observations by BAHAMAS and WALES. GLORIA did not measure during these low level legs, as this part of the flight was mainly inside or just above clouds. At 20:00 UTC, HALO ascended to almost 13 km and performed an east-west leg at 59.5°N (leg 3 from point C to point D). This flight leg was placed further to the south, so GLORIA could look on the earlier performed, lower flight legs (between A and B), which should allow comparisons with in situ and lidar data. 
Unfortunately, the cloud cover prohibited GLORIA during most of the flight leg to collect measurements down to the former flight altitude. At the westernmost point of the leg (point D), HALO went northward back to the original latitude of 61°N (point B) and ascended further to ≈ 13.5 km altitude. A last west-east leg (leg 4 from point B to point A) was performed before returning to the campaign base at Kiruna.

Because jet-generated GWs are not necessarily stationary, linear-flight tomography (LAT) was chosen as GLORIA's mea-
surement strategy, and two separate retrievals were performed using measurements taken during flight leg 3 (southern leg) and flight leg 4 (northern leg), respectively. Both retrievals have a horizontal resolution of 30 km in flight direction and 70 km perpendicular to flight direction. The vertical resolution is 400 m, the precision is better than 0.05 K, and the accuracy, includ-



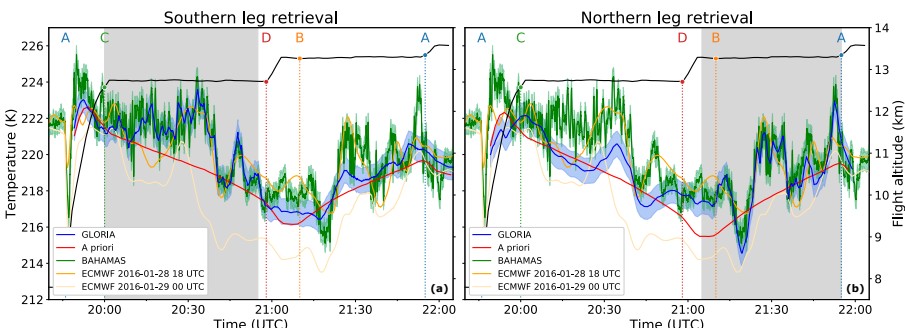

**Figure 5.** A comparison of the GLORIA retrieval results with in-situ-temperature measurements and ECMWF operational analyses. The GLORIA retrievals and ECMWF model data were interpolated in space onto the flight path. The shaded area indicates the time period from which GLORIA measurements were included in the respective retrieval: The southern leg retrieval only uses measurements taken between 20:00 UTC and 20:55 UTC, the northern leg retrieval only those taken between 21:05 UTC and 21:55 UTC. Values in the non-shaded area are extrapolated in space to the earlier or later performed flight path. The black curve shows the flight altitude.

ing misrepresented background gases, uncertainties in spectral line characterization, uncertainties in instrument attitude, and calibration errors, is better than 0.7 K. A detailed description how these retrieval diagnostics are calculated can be found in

Krisch et al. (2018).

The GLORIA southern leg retrieval results agree well with the in-situ temperature measurements of BAHAMAS taken on the southern flight leg (Fig. 5 a between points C and D). The same is valid for the northern leg retrieval results and BAHAMAS measurements from the northern leg (Fig. 5 b between points B and A). Some very small scales are beyond the spatial resolution of GLORIA. In-situ measurements taken during the northern (southern) flight leg, show stronger deviations

when compared to extrapolated GLORIA data from the southern (northern) leg retrieval. However, the main wave structures are still captured. This can be explained by the temporal difference between the two legs and the location of the tangent points of the respective retrievals: The tangent point altitude decreases with distance to the flight path (see Fig. 1). Hence, the tangent points of measurements taken on the southern flight leg are roughly 2.5 km below the flight altitude of 13.5 km of the northern flight leg at 61°N and vice versa. A comparison of in-situ measurements taken for example on the northern flight leg with the

temperature retrieval using measurements from the southern flight leg, thus does not only differ in measurement time but also relies on vertical and/or horizontal data extrapolation. The agreement is still much better than with the a priori temperature.

The large differences between the ECMWF operational analyses at 2016-01-28 18 UTC and 2016-01-29 00 UTC illustrate the high temporal variability of the gravity wave structure. The ECMWF operational analysis at 18 UTC in general agrees very well with the GLORIA and BAHAMAS measurements: it catches the main variations, but the temperature oscil-

lations associated with the GWs are not as detailed as observed by the different measurement techniques. Sometimes the wave structure appears to be shifted in time/space compared to GLORIA and in-situ measurements (e.g., between 20:30 UTC and 20:45 UTC).



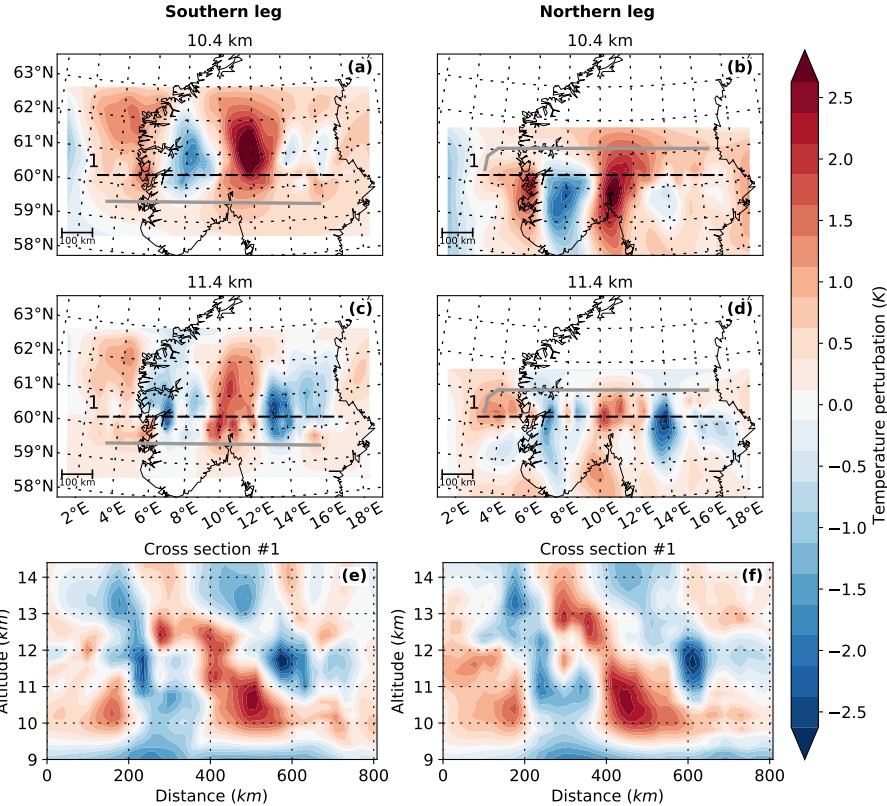

**Figure 6.** Temperature perturbations of the GLORIA tomographic retrieval for the flight on 28 January 2016 over southern Scandinavia. Shown are horizontal (Panels **(a-d)**) and vertical (Panels **(e & f)**) cross sections. The vertical cross sections are along the dashed lines in (Panels **(a-d)**). The grey line indicates the flight path. The left column shows results from measurements taken on the southern flight leg, the right column results from measurements taken on the northern flight leg.

This comparison with both in-situ measurements and ECMWF operational analysis demonstrates the high quality of the tomographic reconstruction of the temperature field from GLORIA measurements and proves LAT using GLORIA capable of reconstructing highly complex gravity wave structures.

## 4 Analysis and discussion

### 4.1 Wave characterization

The temperature retrievals are separated into background atmosphere and GW perturbations using a 3rd order Savitzky-Golay filter with window lengths of 750 km in both horizontal and 5 km in the vertical direction (see Sec. 2.4 and App. A for details). For this filtering, the retrieval data is expanded in all spatial directions with a priori data to avoid edge effects. The remaining





temperature perturbations can be seen in Fig. 6. The left column shows the temperature perturbations derived from the retrieval using measurements taken during the southern flight leg and the right column shows those derived from the retrieval using measurements taken during the northern flight leg.

The GLORIA retrievals for both flight legs show a prominent wave structure with ≈400 km horizontal and ≈6–7 km vertical

wavelength. This large scale gravity wave (LSGW) is perturbed by a smaller scale gravity wave (SSGW) with longer vertical but shorter horizontal wavelength. This SSGW is more prominent in the east at lower altitudes (10.4 km, Fig. 6 a & b) and in the western part at higher altitudes (11.4 km, Fig. 6 c & d). The LSGW has strongest amplitudes of about 3 K between 10°E and 14°E.

Even though the main characteristics are similar for the observations during both legs, there are some differences between

them. The LSGW appears to have slightly different horizontal orientation in the two different retrievals: In the southern leg retrieval between 60°N and 62°N the phase fronts are oriented north-south (Fig. 6 a & c), whereas the phase fronts in the northern leg retrieval seem to be turned slightly and have a north-north-east to south-south-west alignment between 59°N and 60.5°N. Also, the horizontal wavelengths of the LSGW and the steepness of the phase fronts seem to slightly differ between the two retrievals. These differences can either originate from the slight difference in the location of the measurements used

for the two retrievals or the difference in time.

The temperature perturbation fields from both retrievals were spectrally analysed with a 3-D sinusoidal fitting routine in overlapping fitting cubes of 400 km zonal, 250 km meridional, and 4 km altitude extent (see Sec. 2.5 for details). Horizontally, this cube size is of the same order of magnitude as the wavelength. Vertically, the cube roughly encompasses the whole measurement space. To capture the spatial variation of the wave amplitude, refits of amplitude and wave phase, using the

previously determined wave vector $\boldsymbol{k}$, have been performed in smaller sub-cubes of 100 km zonal, 250 km meridional, and 1 km altitude extent.

With these settings, the spectral analysis is only capable of identifying the LSGW component. The results (Fig. 7) confirm the change in horizontal direction of the LSGW between both retrievals observed already in Fig. 6: The wave orientation changes from $\varphi = 270°$ in the southern flight leg retrieval to $\varphi = 290°$ in the northern flight leg retrieval (Fig. 7 d & h). Furthermore,

the horizontal wavelength increases slightly in both retrievals from west to east (Fig. 7 b & f). In the southern leg retrieval, the waves decrease in steepness (decreasing vertical wavelength) from west to east (Fig. 7 c), which can also be seen in the vertical cross section of the temperature perturbations (Fig. 6 e): At 200 km distance along the cross section, the waves have shorter horizontal and longer vertical wavelengths than at 600 km. According to the sinusoidal fit, the LSGW has highest amplitudes between 12°E and 14°E (Fig. 7 a & e). The LSGW in the northern flight leg retrieval is, in general, steeper than those of the

southern flight leg retrieval (Fig. 7 c vs g), a property already visible in the temperature perturbations (Fig. 6).

After the LSGW has been identified in both retrievals, it can be subtracted from the temperature perturbation fields to reveal more clearly the SSGW. The remaining SSGW fields are shown in Fig. 8. Here, SSGWs with amplitudes up to 1.5 K with short horizontal (around 100 km) and very long vertical wavelengths (up to infinity) can be seen. However, the SSGW structure is quite complex and no single monochromatic wave can be identified by eye. Instead, the structure has very localised

maxima and similarity to a chequerboard. This is an indication for the simultaneous presence of at least two wave packets



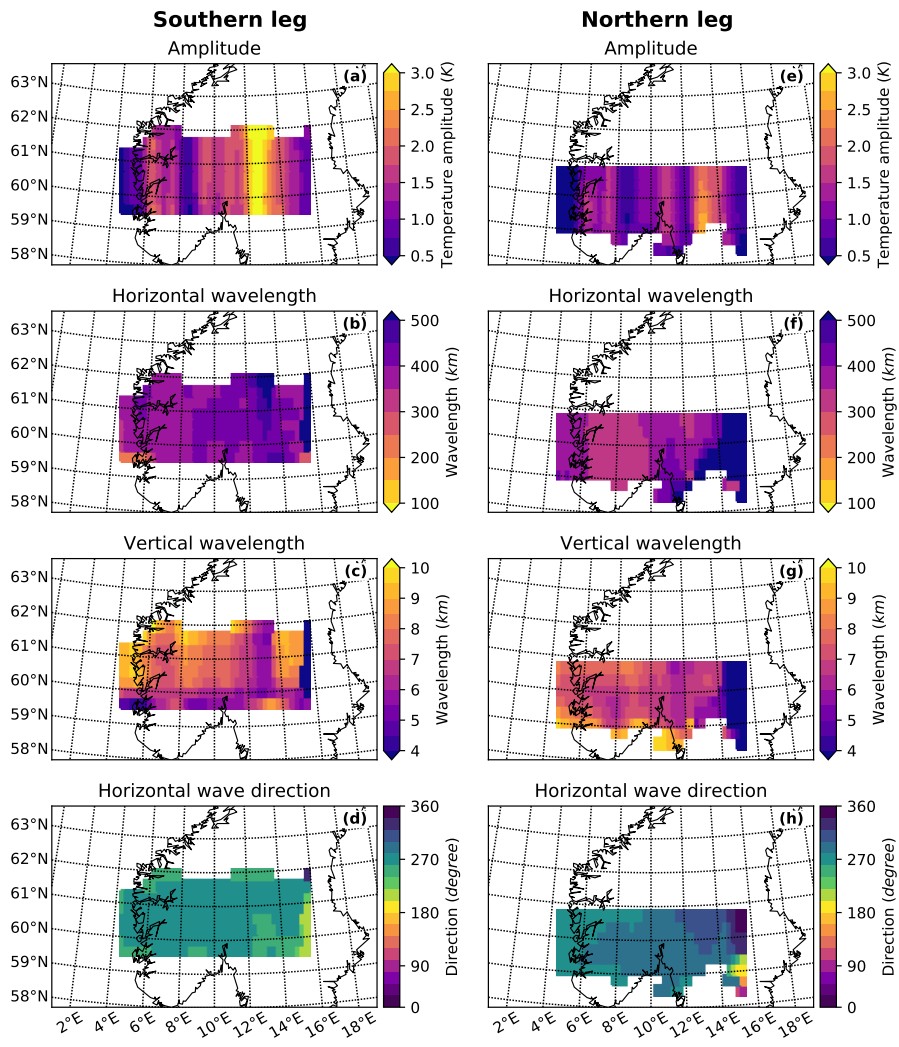

**Figure 7.** Three-dimensional sinusoidal wave fit of the GLORIA measurements at a centre height of $11.4\,\mathrm{km}$ in fitting cubes of 400 x 250 x $4\,\mathrm{km}$ with a tangent point weighting according to Sec. 2.5. In order to capture the spatial variation of the amplitudes, an amplitude and phase refit has been performed in fitting cubes of 100 x 250 x $1\,\mathrm{km}$. Panels **(d & h)** show the direction of the horizontal wave vector. Eastward direction corresponds to $90°$ and southward direction to $180°$





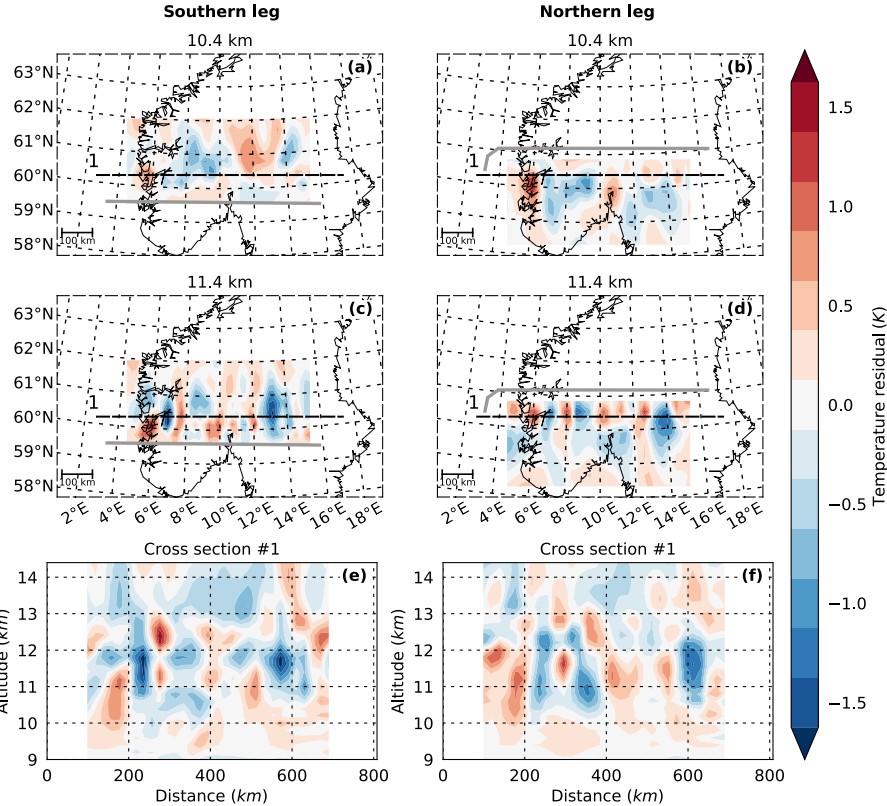

**Figure 8.** Remaining temperature perturbations of the GLORIA tomographic retrieval after subtraction of the wave of Fig. 7. Shown are horizontal (Panels **(a-d)**) and vertical (Panels **(e & f)**) cross sections. The vertical cross sections are along the dashed lines in Panels **(a-d)**. The grey line indicates the flight path.

with different propagation directions and might be caused by the presence of either upward and downward, or eastward and westward propagating waves. There is no indication of a symmetric source, which could explain eastward and westward propagating wave packets. However, simultaneous upward and downward wave propagation might hint to a reflection layer somewhere above the measurement altitude.

Not only the LSGW component changes in time: The phases of the SSGW component shifts a bit further to the east around an altitude of 11.4 km from southern to northern leg retrieval (Fig. 8). This can be seen, for example at the maximum at 8°E which is located slightly to the left of the meridian for the southern retrieval, whereas it is on the meridian for the northern retrieval. The two maxima between 10°E and 11°E show a similar behaviour. All these differences between two retrievals explain, why a joint retrieval using measurements of both legs simultaneously did not converge properly.

As the sinusoidal fitting routine currently is only tested for fits of one monochromatic wave at a time, chequerboard patterns cannot be resolved. To spectrally analyse the observed SSGW field with the method described in Sec. 2.5, the fitting routine would have to be further tested and potentially adjusted. This is beyond the scope of this paper.





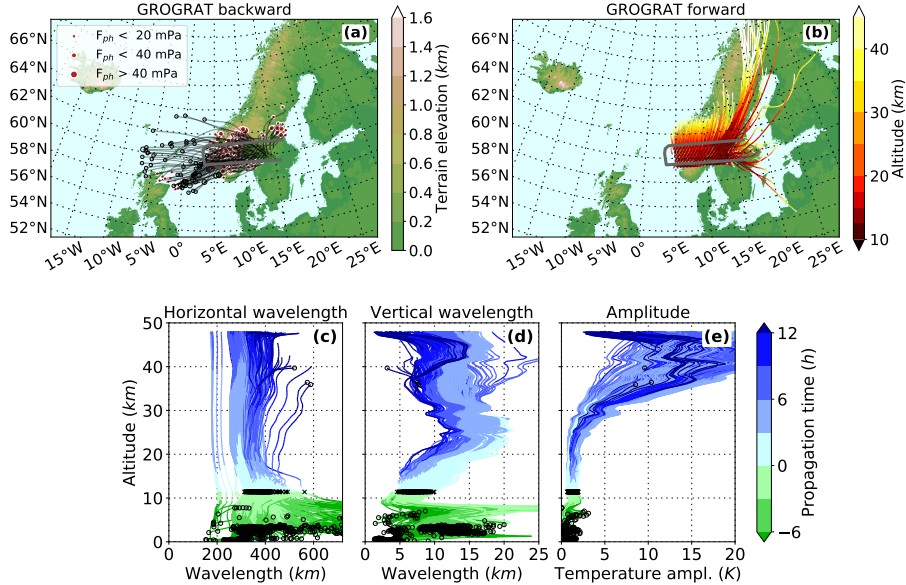

**Figure 9.** GW ray traces calculated using the GROGRAT model. Panel **(a)** shows the backward ray traces and Panel **(b)** the forward ray traces. Panels **(c-e)** show the change of wave parameters with height. The end points of backward rays which do not reach the surface are marked with an open circle, rays which reach the surface are indicated with a red dot. The size of the circle marks the strength of the wave (GWMF). In Panels **(c-e)**, the crosses indicate which wave properties the waves, which do not reach the surface, have at flight altitude. However, no obvious pattern can be observed.

## 4.2 Wave sources and propagation

In order to identify the sources of the LSGW component, ray-tracing calculations with GROGRAT have been performed

(Sec. 2.6). Such ray-tracing calculations need very accurate GW starting parameters (cf. Appendix of Krisch et al., 2017), which could be obtained by the sinusoidal fit only for the LSGW component. Thus, only the sources of the LSGW are analysed in the following.

Most of the backward rays of the LSGW component and especially those with highest gravity wave momentum flux (GWMF) values are traced back to the Scandes (Fig. 9 a). However, other rays and especially those not reaching the sur-

face originate from a widespread area west of Scandinavia. According to ERA5 (Sec. 3.2), a jet-exit region as well as a low pressure system were moving over this area during the course of the 28 January 2016. Both might be the source of these non-orographic GWs. At the measurement altitude, the wave parameters of waves not originating from the surface (Fig. 9 c-e black crosses) do not differ significantly from those generated by orography.

The sources of these waves are further examined by comparing the ray-tracing results with ERA5 (Fig. 10). One ray trace has

been chosen as a non-orographic GW reference case and ERA5 cross sections are plotted along its path. The wave source can be located at any point along the backward trajectory of the ray-tracer. In the early morning at 03:00 UTC, the GW predicted





**Figure 10.** Cross sections through different ERA5 temperature perturbations along an exemplary GROGRAT ray trace originating from GLORIA measurements. The left column shows vertical cross sections along the ray trace (black line). The grey dot marks the location of the ray trace at the respective time step of the model. The green line shows the orientation of the phase lines as predicted by GROGRAT. The right column shows horizontal cross sections at the altitude of the ray path at the respective model time.





by the ray tracer does not agree well with ERA5. Thus, the source of the wave might be further towards the measurement location. At 09:00 UTC a wave structure with similar orientation as the one predicted by the ray tracer can be found just in front of the Scandinavian coast in ERA5. However, the orientation of the wave is not aligned with the main mountain ridge.

Moreover, the location of the wave is still off the coast and not above the mountain range. Both elements suggest an excitation by a non-orographic source.

At 15:00 UTC, a wave field located directly above the mountains and reaching up to 20 km appears in ERA5. However, the wave structure at 10 km altitude, i.e. the exact location of the traced wave, differs in steepness from the fields above and below. At 20:00 UTC, the time of the measurement flight, this slightly flatter structure has propagated a bit further. In the horizontal

cross section, the cold front (blue) has an orientation more or less parallel to the main mountain ridge south of $62°$N. North of $62°$N, the orientation changes and agrees well with the prediction of the ray tracer. At 02:00 UTC on the following day, one can now clearly identify different wave packets both in the horizontal as well as in the vertical cross section. The wave packet followed by the ray tracer is less steep than the waves above the mountains and is now located further to the east. This comparison suggests, that a non-orographic wave packet has travelled through an orographically excited wave above the

Scandes during the course of the late afternoon and night of the measurement day. This again explains why the retrieval of both flight legs simultaneously did not converge: the temperature perturbations caused by the non-orographic wave were not sufficiently stationary.

Forward ray tracing shows, that the waves propagate slightly northward and to high altitudes (Fig. 9 b). The temperature amplitude increases with height and reaches values between 10–30 K just below 40 km. The waves take between 3–12 h to

propagate to these altitudes. The exact propagation time strongly depends on the wavelength: gravity waves with long vertical and short horizontal wavelengths (steep waves) rise faster than those with shorter vertical and longer horizontal wavelengths. The horizontal wavelengths stay on the order of 200–400 km. The vertical wavelengths double from 5–10 km at GLORIA measurement altitude to around 10–20 km at an altitude of 20 km and stay more or less constant above. This doubling of the vertical wavelengths is the result of a Doppler shifting caused by a doubling of the horizontal wind from $30\,\mathrm{m\,s^{-1}}$ at 12 km to

$60\,\mathrm{m\,s^{-1}}$ above 20 km altitude (Fig. 3).

### 4.3 Comparison to AIRS measurements

To investigate, how accurate the forward ray-tracing calculations of the GROGRAT model are, the propagation results are compared to AIRS satellite measurements. GROGRAT predicts the GWs to take between 3–12 h to propagate from GLORIA measurement altitudes up to 36 km. Thus, AIRS measurements of the descending orbit on 29 January 2016 were chosen for the

comparison (Fig. 11). These measurements over Scandinavia were taken between 01:00 UTC and 03:00 UTC that is between 3 h and 6 h after the HALO flight took place. The forward ray tracing predicts GW amplitudes between 10 K and 30 K above middle and northern Scandinavia (Fig. 9 e). The vertical wavelengths are predicted to be between 10 km and 20 km (Fig. 9 d). According to the AIRS sensitivity function (Fig. 2) such GWs are underestimated in amplitude by roughly 80 % and overestimated in vertical wavelength by around 20 %. Thus, these waves should appear only weakly in the AIRS measurements and

with wavelengths around 18 km. This is confirmed by the AIRS temperature perturbations at 27 km and 36 km (Fig. 11): Above





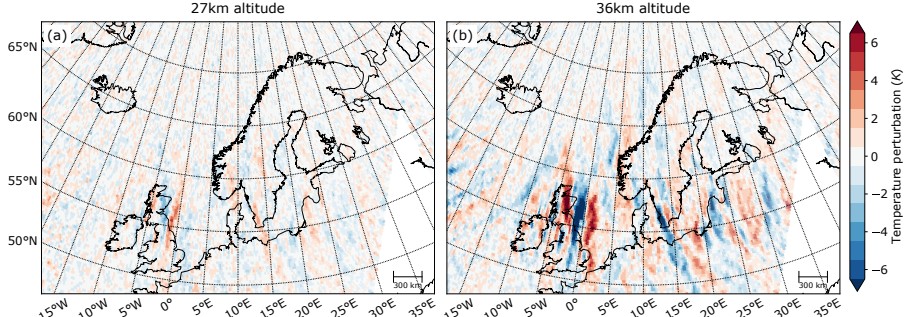

**Figure 11.** Temperature perturbations of the AIRS retrieval at 27 km (a) and 36 km (b) for the descending orbits with equator crossing time at 01:30 LT (between 01:00 UTC and 03:00 UTC above Scandinavia) on 29 January 2016.

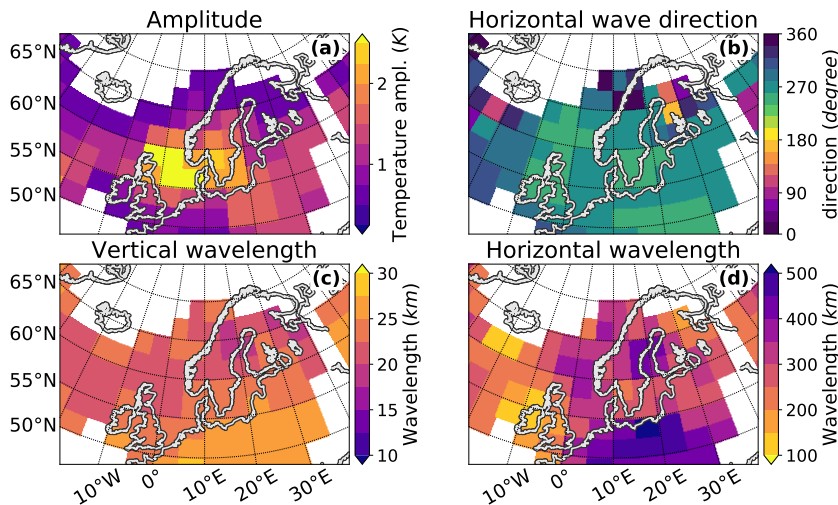

**Figure 12.** Three-dimensional sinusoidal wave fit of the AIRS measurements based on fitting cubes of 300 x 250 x 20 km at a centre height of 36 km.

60°N barely any temperature perturbations are visible. Sinusoidal fits of the AIRS data in cubes reaching from 26 km to 46 km show high amplitudes above the southern tip of Scandinavia and the North Sea (Fig. 12), where the mid-stratosphere wind velocities are higher (cf. Fig. 3 f). Above middle and northern Scandinavia, as expected, very low amplitudes are identified with vertical wavelengths on the order of 20 km. Furthermore, the horizontal wavelengths derived from the AIRS measurements

comply well with the GROGRAT model results.

The influence of the AIRS sensitivity on these GWs is studied in more detail using ERA5 model data. The ERA5 temperature field is first separated into small scale gravity wave perturbations and large scale background motion (see Sec. 2.4). Each profile of the ERA5 GW perturbation field is then multiplied with the AIRS averaging kernel matrix. The results are shown in Fig. 13. At an altitude of 27 km the ERA5 field is filled with various GWs of amplitudes on the order of 3 K (Fig. 13 a). After applying

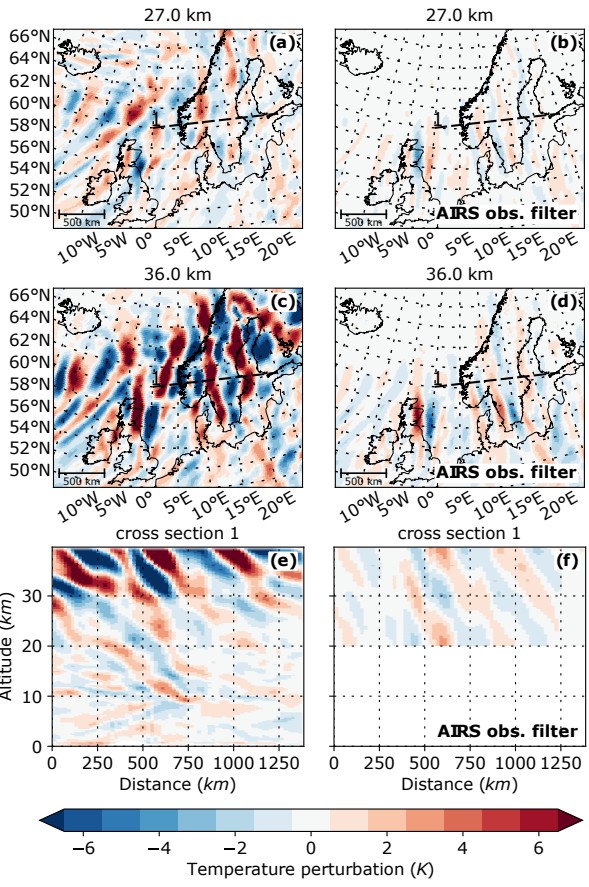

**Figure 13.** ECMWF forecast initialized on 29 January 2016 midnight for 03:00 UTC and the influence of the AIRS observational filter. The left column shows the original ECMWF forecast data, the right column shows what remains if the model data is multiplied with the averaging kernel matrix of the AIRS retrieval.

the AIRS averaging kernel, only small parts of the wave structure remain visible with strongly damped amplitudes (Fig. 13 b). Also the complex wave structures are replaced by mainly monochromatic wave packets. A similar picture can be seen at 36 km altitude (Fig.s 13 c & d). In addition to this amplitude underestimation, the vertical cross sections reveal the overestimation of the vertical wavelengths (Fig.s 13 e & f), which had already been predicted by the sensitivity function in Fig. 2. In particular, the flat waves on the top right of Fig. 13 e with vertical wavelengths on the order of 10 km appear with very low amplitudes

and much steeper phase fronts in the AIRS simulation (Fig. 13 f). A similar overestimation of vertical wavelengths by AIRS was also observed by Meyer et al. (2018) for a strong wave event over South America, when comparing AIRS measurements to those of the limb sounder HIRDLS which has a much better vertical resolution.

A comparison of these simulated AIRS measurements (Fig. 13 d) with the real AIRS measurements (Fig. 11) shows an excellent agreement. However, due to the different visibility filters of the measurement techniques, the GWs observed by





GLORIA at lower altitudes and propagated forward by GROGRAT are only barely visible for AIRS. GLORIA and AIRS cover rather different parts of the full gravity wave spectrum.

## 5 Conclusions

In this paper, a complex gravity wave field above southern Scandinavia was examined with respect to its sources and propagation paths. Measurements taken with GLORIA on the 28 January 2016 on two consecutive linear flight legs show a complex
wave field, composed of multiple wave packets with different spatial structure, demonstrating the capability of GLORIA limited angle tomography to reproduce complex wave patterns. Even though the overall wave structure is similar in both retrievals (one from each flight leg), some difference in wave orientation and the location of small features can be seen. These differences stem from the slight difference in space and time.

A three-dimensional spectral analysis revealed large scale waves with horizontal wavelengths around 400 km and vertical
wavelengths between 5 km and 7 km. The different vertical wavelengths originate from multiple wave packets in the same analysis field. The different large-scale wave packets were distinguished and characterised by the S3D spectral analysis method.

After subtraction of the large-scale waves, a very complex small-scale wave field with a chequerboard structure remained. Such a chequerboard pattern is an indication of a superposition of at least two wave packets with different propagation directions. To distinguish and characterise these small-scale wave packets improved S3D fits would be required.

The large-scale wave components were analysed further with the GROGRAT ray tracer and three potential sources were identified: the orography of the Scandes and both a jet-exit region as well as a low pressure system, which were travelling from west to east over the Atlantic Ocean and southern Scandinavia. The ray traces going back to the orography propagate almost vertically upwards through the GLORIA measurement volume and up into the mid-stratosphere, while the backward ray traces not reaching the mountains originate from west of the Scandinavian peninsula and cross the mountain wave region from west
to east exactly at the GLORIA measurement altitude. Therefore, not only the small-scale wave component consists of multiple wave packets, but the large-scale wave component, too.

A comparison of one ray trace with ERA5 model data, confirms the prediction of two wave packets crossing each other. According to both models, GROGRAT and ERA5, the two wave packets propagate up to the middle stratosphere. However, due to the limited measurement sensitivity of AIRS to vertically small-scale GWs, the stratospheric satellite measurements
strongly underestimate the wave amplitudes and overestimate the vertical wavelengths. The remaining wave signal in the AIRS measurements agrees qualitatively very well with the predictions by the ray tracer and ERA5. For an exact quantitative comparison either another satellite instrument with higher vertical resolution or a gravity wave with longer vertical wavelengths in the stratosphere would have been required.

In summary, this study demonstrated that limited-angle tomography using GLORIA is a well-suited tool to observe complex
gravity wave fields in 3-D in the UT/LS region and accurately identify several wave components simultaneously. At the same time, such highly resolved 3-D observations challenge the currently existing analysing techniques, e.g. S3D, which will have to be expanded to describe gravity wave interference patterns such as chequerboard patterns in the future. Furthermore, the





accuracy of forward and backward ray-tracing shown in this study opens new possibilities for combining ray-tracing with dedicated 3-D measurements in even more complex situations to gain a better understanding of gravity wave sources and
propagation patterns. Last but not least, the example case shows that even in the presence of a prominent mountain ridge the observed wave patterns can be determined from different sources of comparable strength.

*Data availability.* The tomographic retrieval data of GLORIA is available from the HALO database (Krisch and Ungermann, 2020a, b). AIRS retrieval data are available by contacting L. Hoffmann, Forschungszentrum Jülich. The ECMWF analysis are available directly through ECMWF (ECMWF, 2017), ERA5 fields are available through the Copernicus Climate Data Store (C3S, 2017).

*Competing interests.* The authors declare that they have no conflict of interest.

*Acknowledgements.* This work was partly supported by the Bundesministerium für Bildung und Forschung (BMBF) under projects 01LG1206B and 01LG1206C (ROMIC/GW-LCYCLE), as well as by the European Space Agency (ESA) under contract 4000115111/15/NL/FF/ah (GWEX) and the Deutsche Forschungsgemeinschaft (DFG) project ER 474/4-2 (MS-GWaves/SV), which is part of the DFG researchers group FOR 1898 (MS-GWaves). The authors gratefully acknowledge the computing time granted through JARA on the supercomputer JU-
RECA at Forschungszentrum Jülich (Jülich Supercomputing Centre, 2018). We sincerely thank A. Dudhia, Oxford University, for providing the Reference Forward Model (RFM) used to calculate the optical path and extinction cross-section tables required by our forward models. D. E. Kinnision, NCAR, is thanked for kindly providing the WACCM4 model data used in the retrieval. The European Centre for Medium-Range Weather Forecasts (ECMWF) is acknowledged for meteorological data support. The results are based on the efforts of all members of the GLORIA team, including the technology institutes ZEA-1 and ZEA-2 at Forschungszentrum Jülich and the Institute for Data Process-
ing and Electronics at the Karlsruhe Institute of Technology. We would also like to thank the pilots and ground-support team at the Flight Experiments facility of the Deutsches Zentrum für Luft- und Raumfahrt (DLR-FX).

## Appendix A: Comparison of different scale separation methods for GLORIA measurements

Due to the local nature of GLORIA measurements, global filtering algorithms, as used for model data and satellite instruments, are not suitable for the scale separation of the atmospheric temperature. Furthermore, GLORIA measurements do not have the
same spherical latitude-longitude grid as model data. Instead they are sampled to regular Cartesian coordinate systems with km-distance to a reference point as x- and y-coordinates. The reference point is chosen ad-hoc for each retrieval separately and is always located somewhere in the centre of the measurement volume. A number of low-pass filters are suitable for the scale separation on regional data sets. To identify the best method for the GLORIA measurements, a 2-D FFT-filter, a running mean filter, a Gaussian filter, an SG-filter, and a Butterworth filter (BW-filter; Butterworth, 1930) are compared in the following.
The separation of pass and stop frequencies are handled differently in each method (Fig. A1). The FFT-filter has a very sharp





**Table A1.** Different filters used for the scale separation of GWs and background and their set-up parameters.

|  | polynomial | cut-off wavelength | window length | FWHM |
|---|---|---|---|---|
| Fast Fourier transform (FFT) |  | 750 km |  |  |
| Running mean |  |  | 750 km |  |
| Gaussian |  |  |  | 500 km |
| Savitzky-Golay (SG) | 3rd order |  | 750 km |  |
| Butterworth (BW) | 3rd order | 750 km |  |  |

transition from pass to stop band, but requires a periodic signal, which GLORIA measurements cannot provide. Assuming the GLORIA measurements to be periodic in space, introduces edges effects as can be seen in Fig. A2 g-i. The running mean filter and the Gaussian filter have both a very flat transition between pass and stop band. This makes a clear separation more challenging. In contrast, the SG-filter as well as the BW-filter have a faster transition between pass and stop band.

To test these filters systematically on GLORIA-like data, a synthetic temperature field is constructed, which covers an altitude range from 8-15 km and has a horizontal extent of 1000 km centred around the coordinate origin (Fig. A2 d-f). This temperature field is composed by a superposition of an international standard atmosphere profile (ISO 2533:1975), a synoptic scale wave and a mesoscale GW (Fig. A2 a-c). The international standard atmosphere is defined in two altitude ranges: Above 11 km, a constant value of 216.15 K is assumed; below 11 km altitude, the temperature decreases with a constant gradient of 490  -6.5 K km$^{-1}$. As the filtering methods are very sensitive to abrupt changes, a running mean with a 1 km window is applied to the standard atmosphere profile to smooth the transition between the two regimes. The synoptic scale wave has a wavelength of 1500 km (corresponds to wave number 12 at 60° latitude), phase fronts oriented parallel to the y-axis and a temperature amplitude of 1.5 K. The mesoscale GW is chosen to have a horizontal orientation perpendicular to the synoptic scale wave, a horizontal wavelength of 300 km, and a vertical wavelength of 5 km. The constructed wave is further multiplied by Gaussian 495  functions in all spatial dimensions to simulate the often localised nature of real GW packets. The Gaussian functions have a FWHM of 400 km in both horizontal directions and a FWHM of 5 km in the vertical. The sum of mean temperature, synoptic scale wave and GW (Fig. A2 d-f) is used as input for the different filtering algorithms.

All filtering algorithms are applied sequentially in both horizontal dimensions to avoid that GWs which are oriented along one horizontal axis are erroneously considered as background. The exact set-ups of the different filters are summarized in 500  Table A1. The results are shown in Fig. A2. With the FFT-filter (third row), the running mean (fourth row) and the Gaussian-filter (fifth row), parts of the synoptic scale wave remain in the perturbation field. Thus, these filters are not appropriate for the scale separation of GLORIA data. Both, the SG-filter (sixth row) as well as the BW-filter (seventh row) qualitatively reproduce the original GW structure (Fig. A2 a-c) with minimal altering effects. The BW-filter seems to shift the wave phases outwards, which is likely to be due to a small part of the synoptic scale wave remaining in the signal. A quantitative comparison is done





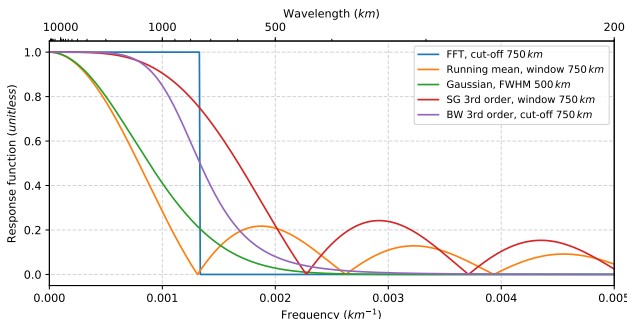

**Figure A1.** Frequency response of different low-pass filters to a delta function in spatial space. Shown are a Fast Fourier Transform (FFT) with a cut-off wavelength of 750 km, a running mean filter with a window width of 750 km, a Gaussian filter with a full width at half maximum (FWHM) of 500 km, a Savitzky-Golay (SG) 3rd order polynomial smoothing in running windows of 750 km width, and a 3rd order Butterworth (BW) filter with a cut-off wavelength of 750 km.

by calculating the Pearson coefficient $P$ correlating the original wave with the filtered results:

$$P = \frac{\sum_{i=1}^{n} (x_i - \bar{x})(y_i - \bar{y})}{\sqrt{\sum_{i=1}^{n} (x_i - \bar{x})^2} \sqrt{\sum_{i=1}^{n} (y_i - \bar{y})^2}}, \tag{A1}$$

with $x_1 \ldots x_n$ all data points of the original wave field, $\bar{x}$ the mean of the original wave field, $y_1 \ldots y_n$ all data points of the remaining wave field after filtering, and $\bar{y}$ the mean of the remaining wave field after filtering. The FFT-filter reaches a correlation with the original of 53.2%, the running mean of 51.5%, the Gaussian of 86.9%, the SG-filter of 99.4% and the BW-

filter of 98.5%. Thus, the Pearson coefficients confirm that the SG-filter is the best choice for GLORIA-like measurements. Other orientations and wavelengths of both synoptic scale waves and GWs have been tested and lead to similar results.

     Including an additional filter over the altitude dimension can further help to remove the effects of small scale weather systems. Thus, for the GLORIA measurements presented in this paper, an additional 3rd order SG-filter with a window length of 3 km is applied in the vertical after the horizontal filtering.



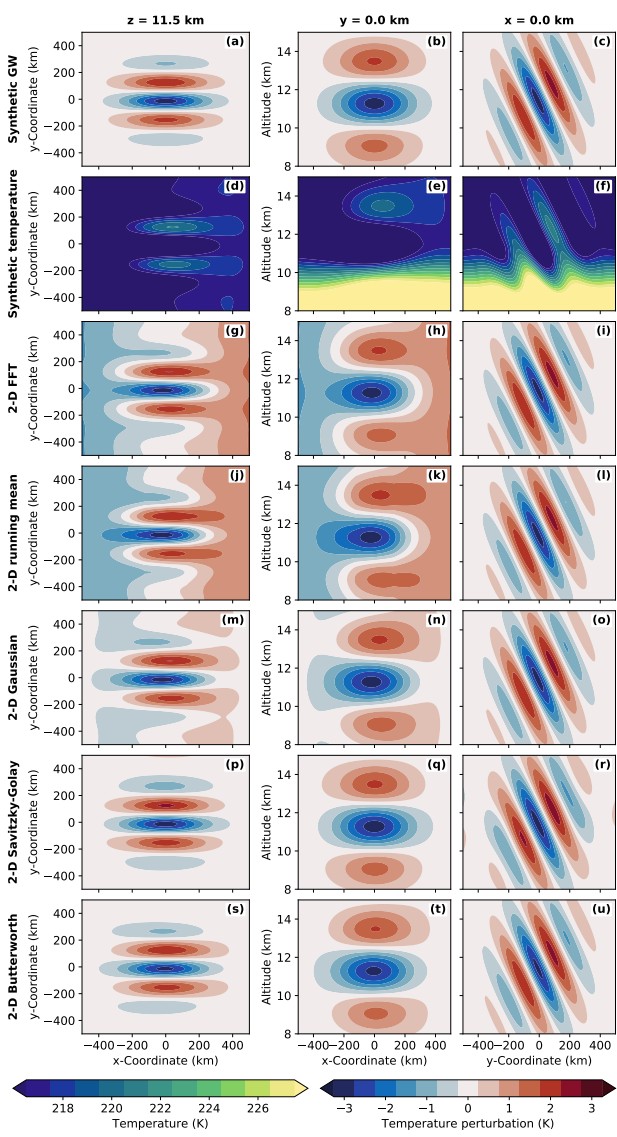

**Figure A2.** Comparison of different scale separation methods applied to a synthetic temperature field. The left column shows horizontal cross sections at 11.5 km altitude, the middle column cross sections in the x-z-plane along the y-axis, and the right column cross sections in the y-z-plane along the x-axis. The synthetic temperature (d-f) is constructed from the international standard atmosphere (ISO 2533:1975), a synoptic scale zonal wave, and a mesoscale GW (a-c). Detailed descriptions of the different fields and their exact structure can be found in the text. Temperature fluctuations calculated by subtracting the low-pass filtered background fields from the original synthetic temperature field for different filtering techniques are shown on rows 2-6. A perfect filter should be able to fully reproduce the synthetic GW structure shown in the first row.



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
