# Peer review of "Superposition of gravity waves with different propagation characteristics observed by airborne and space-borne infrared sounders"

_Atmospheric Chemistry and Physics, 2020_

## Referee Comment (RC1) · Neil Hindley (Referee) · 22 Jun 2020

**Neil Hindley (Referee)**

n.hindley@bath.ac.uk

Received and published: 22 June 2020

**Overview**

In this study the authors analyse observations of gravity wave structures over Scandinavia using measurements from the airborne GLORIA instrument. They apply 3-D spectral analysis to measure several overlapping waves in the GLORIA measurements, then ray-trace these waves forwards and backwards in time to discover their sources and trajectories. By using reanalysis data to provide the synoptic background fields, they are able to identify both probable mountain waves and non-orographic waves.

They also analyse waves in co-incident satellite measurements from AIRS/Aqua and ERA5 reanalysis which support their findings.

Overall I find that this paper, the data, the analysis methods and the results are of a very high standard and I believe it should be accepted into ACP. It falls well within the scope of the journal. It is always nice to read a paper that describes measurements from a field campaign, rather than simply a modelling or satellite study, and this field campaign involved some very cutting-edge equipment and analysis. The results will be of use to the community not only for science purposes but also for guiding the directions of future airborne campaigns.

I have a range of only minor comments listed below, mainly asking for clarity on how the authors report their results and describe their methods.

I have also included a Minor Suggestions section, which describes some minor grammatical/typographic changes and some optional suggestions. I do not require a pointby-point response to the Minor Suggestions section.

**Minor Comments**

1. **Abstract**, **I.12-13** The phrasing is a bit strange here. If the AIRS measurements barely resolve the wave, how can they support the results? The word "barely" is usually used in a negative context to dismiss something rather than use it to support something else, so I'd suggest rephrasing to:

"Co-located AIRS measurements in the middle stratosphere are also in agreement with these results, despite their coarser vertical resolution compared to GLORIA measurements."

2. **I.28** If relevant, I think the authors might like to include Vosper and Ross (2020, doi:10.3390/atmos11010057) here, which has some important considerations for

gravity wave momentum flux measurements from near-vertical profiles.

- 3. **I.47** The 3-D *S*-transform method is fully described, tested and validated in Hindley et al (2019, doi.org/10.5194/acp-19-15377-2019) so this would be a useful reference to included here. This is the planned 'method paper' than underlines the 3DST approach we applied in Wright et al (2017), it just took me a little longer to finish it!
- 4. Introduction The introduction could benefit from a brief synopsis of the paper at the end, since the next section is quite technical. In particular, it would be nice to describe the aircraft campaign a little before hitting the reader with a wall of technical detail. Perhaps something like

"Here we analyse airborne GLORIA measurements from a flight over Scandinavia in 2016. We apply ray-tracing techniques to our measurements and compare our results to satellite observations and reanalysis. The datasets, spectral analysis and ray-tracing methods are described in Sect. 2. The flight campaign and synoptic conditions are described in Sect. 3... etc... "

- 5. **Fig. 1** The figure is nice but the caption is quite confusing. I would perhaps suggest to start with: "Vertical (a) and horizontal (b) cross-sections of the limb sounding geometry of airborne GLORIA measurements using the LAT configuration. Measurements are made at the tangent points shown by the coloured dots, where the colour indicates the tangent point altitude. In panel (a), ... "
- 6. Fig. 1 "Images taken under 90° azimuth cover the dark grey area with the LOS" this sentence does not make sense, please rephrase.
- 7. **Fig. 2** The authors should make it clearer how this AIRS sensitivity function is derived, in particular how the effect on measured vertical wavelength is found. Was this function derived mathematically, or were synthetic waves created, then passed through the AIRS vertical retrieval, then measured with the 1-D fitting

routine they describe to compare vertical wavelengths in/out? Is this the same approach as the correction factor applied in the supporting information of Ern et al. (2017)?

Either way, I think it is important to briefly state how this is derived. It's not made clear in the text, or in the references given, how the stretching of the vertical wavelength takes place mathematically (apologies if I have missed it in the references to Meyer et al (2018) and Ern et al (2017)). Also, is this stretching the same for all altitudes in the AIRS retrieval?

I am confident that the authors are correct, but it needs to be made clearer in order to convince the impartial reader. Only one or two sentences are required.

8. **I.118-120** Looking again at Fig. 2, I think the threshold should be 25km because above this the waves are slightly shrunk, which would introduce a low bias in GWMF results.

AIRS is not the focus of this study, but for information I find that in practice the vast majority of vertical wavelengths measured in AIRS are generally between 15 and 25km (e.g. Hindley 2019, doi.org/10.5194/acp-19-15377-2019). Shorter vertical wavelengths tend to be lost below the noise threshold, and longer vertical wavelengths tend to be underestimated due to only a few cycles being present in the vertical, which reduces the signal to noise. We did not apply a correction in the measured vertical wavelength in Hindley et al (2019) because I found that the measurement error in the vertical wavelength of around 10-25% was comparable to the correction factor, so I didn't want to incorrectly apply it and introduce a further source of error. Here, error in the vertical wavelength measurement of the S3D method could be similar, so they may have the same problem, but the authors do not discuss or quantify that here. Anyway, AIRS is not the focus of the paper. Perhaps, to avoid further discussion, the last two sentences could be changed to simply:

"For vertical wavelengths below 25km, the temperature amplitude is significantly underestimated and measured vertical wavelengths in AIRS can appear up to 45% larger than their true value. As such, AIRS measurements of vertical wavelengths below 25km may be over-estimated, so caution is advised."

9. **I.140** I assume the authors are describing operational analysis here, as distinct from reanalysis? Including the term "operational analysis" would make this clearer. Suggest rephrasing the line to

"The ECMWF operational analysis for the year 2016 used here uses...", if that is what is meant.

- 10. **I.203** I think scale height H and buoyancy frequency N still need to be defined.
- 11. I.215 It's interesting that the authors used the ECMWF operational analysis for the GROGRAT ray-tracing and not the ERA5 reanalysis. It's a bit strange to interpolate the 6-hour operational analysis in time when ERA5 reanalysis is available hourly, which has improved accuracy anyway due to more data assimilation. I don't think it affects the results, but it is a limitation. Was it needed to be run in real time during the campaign? If I have missed something, I apologise.
- 12. **I.240** It would be good to include the equation used here to find this number, if possible.
- 13. **Fig. 3** The coastline is very difficult to see in grey. Perhaps either apply the same coastline method as in Fig. 12, or maybe the coastline in black and the wind vectors in a bright blue or something? If this is a lot of work, do not worry, the figure is acceptable as it is.
- 14. **I.341** Are eastward waves permitted here in the background wind conditions? Given that they must be reasonably slowly moving in order to be measured by GLORIA?

15. **I.352** Perhaps change "This is beyond the scope of this paper" to "These results provide guidelines for the future development of the S3D method." - Turn a negative into a positive :)

Out of interest, how much has the S3D code changed between Lehmann et al (2012) to Ern et al (2017) to Krisch et al (2017) to now?

- Fig. 12 The largest wave amplitudes measured in the S3D cubes seems to be shifted to the north-east of where the largest wave perturbations are found in Fig. 11b, between +/- 5 degrees longitude. Is there a reason for this or just where the cubes lie?
- 17. **Fig. 12** The authors should briefly discuss the apparent reduction in the S3Dmeasured amplitude in Fig. 12a compared to the temperature perturbation amplitudes in Fig. 11b to the east of Scotland. I am aware that the S3D method underestimates wave amplitudes due to averaging over the cube size but it should be mentioned here too if that is what is going on. Sorry if I have missed it.
- 18. I.400-401 Again, the use of "barely" here is not very useful. The panels in Fig. 11 show measurements that are around 9km apart in the vertical. This is slightly more than one atmospheric scale height. Due to the exponential increase in wave amplitude with height due to decreasing density, one might expect that a gravity wave's amplitude would increase by perhaps a factor of 2 or more over this height range. If the colour limits on Fig. 11a were set to +/- 3.5K, rather than around +/- 7K as in Fig. 11b, then I think many more temperature perturbations would be "visible". The problem is, however, that the desired gravity wave temperature perturbations are too small to be visible above the retrieval noise, as the authors discussed above, due to amplitude attenuation for shorter vertical wavelengths.

The authors should perhaps rephrase the sentence to reflect this. If it is straightforward to do, the colour limits of Fig. 11a could also be changed and given its own colour bar to make it more useful. 19. I.408 Could the authors provide a reference to the exact averaging kernel that was applied here? Also, I don't think a simple multiplication operation is sufficient. I think either the full AIRS vertical retrieval should be run on the ERA5 fields or, at the very least, each height level should be separately convolved with the appropriate averaging kernel for that height and then combined. One cannot apply a resolution kernel just by multiplication alone, I believe convolution is needed (unless that multiplication is in the Fourier domain of course, in which case it is convolution). Sorry if I have misunderstood this from what the authors have written.

I do have confidence that the authors have used an appropriate method, but they should describe it more clearly. Neither the exact kernel nor the method is shown anywhere in the paper, or in the references listed (sorry if I have missed it), nor how the stretching of the vertical wavelength that occurs in AIRS measurements arises from the maths. Do they mean the kernels shown in Hoffmann and Alexander (2009)? If so, how are they combined?

By the way, I am aware of the current condition of Lars Hoffmann, so if this step was performed by him and the authors cannot ask him then do not worry, a reference the relevant kernels in Hoffmann and Alexander (2009) will be sufficient.

- 20. Fig. 13 caption See above point about multiplication.
- 21. **I.413** As mentioned above, it would be good to have more information on how the sensitivity function is calculated.
- 22. **I.420-421** "GLORIA and AIRS cover rather different parts of the full gravity wave spectrum". If the authors make a revision of the text in Sect. 4.3, I think this should be more of the main theme running through the discussion of results. It could be clearer that, as I understand it, the story is that the authors measure the waves in GLORIA, show that they propagate upwards via forward ray-tracing and are resolved in AIRS data above. Despite the reduced vertical resolution

of AIRS, and the inherent amplitude/wavelength errors discussed, the wave is still resolved at 36km, showing that their GLORIA measurements and ray-tracing results are valid. This is a success. The ERA5-as-AIRS data also shows this, which is a further success. Instead, the language used in the AIRS comparison is somewhat defensive and it does not need to be. The two instruments measure different parts of the GW spectrum, they both have their own limitations as with all observations, and here they measure at two different altitudes. As I mentioned above regarding the abstract, I believe the key result here is that the waves are resolved above GLORIA almost exactly as expected *despite* the resolution difficulties of the AIRS measurements. Again, this presents the results in a positive light rather than being lost in technicalities.

Anyway, to me this just shows that the authors have a good attention to detail, so I leave it up to them if they want to adjust the text.

23. **I.443-446** See comment above. Here, I would suggest that the order the two sentences is reversed. This presents the results positively:

"Gravity wave signals in AIRS measurements agree qualitatively very well with the predictions by the ray tracer and ERA5. This is despite a strong underestimation of wave amplitudes for waves with vertical wavelengths shorter than around 25km. Furthermore, we report an overestimation of the vertical wavelengths for AIRS measurements."

However, I would suggest not including the last sentence I wrote above unless the authors have shown (or more clearly referenced) how the vertical wavelength underestimation occurs in AIRS measurements.

24. **Appendix A** I just want to mention that the inclusion of Appendix A is very good, particularly the frequency response figures for different filters for GW analysis. Many studies simply choose a method and plot the results, so it is very important that filtering choices here are properly assessed, as the authors have done here.

25. **I.510** Is there a reason the authors chose the SG-filter over the Butterworth? The SG-filter has quite a bit of unwanted "ringing" in the pass-band, which the Butterworth does not. However I can see in Fig. A2 that the Butterworth does some strange "shifting" of the 2-D perturbations which the SG-filter does not.

**Minor Suggestions**

- 1. **Abstract** Perhaps the authors might like to add one brief sentence on the motivation of the study, and one on the significance of the results, for readers in adjacent fields?
- 2. Abstract Perhaps make it all one paragraph?
- 3. **Abstract** All gravity waves *are found to* propagate upward into the middle stratosphere.
- 4. **I.26** It might be nice to list the measurement techniques too rather than just their products I assume here in this sentence the authors are talking about radiosonde measurements and others?
- 5. **I.38** delete "nicely"
- 6. **I.39-40** Suggest change sentence to: "The measurement technique of the GLO-RIA instrument and subsequent data processing are described in Section 2."
- 7. **I.56** ... in Section 4.
- 8. **I.59** The GLORIA acronym is already defined, so perhaps just say "The airborne GLORIA instrument measures the infrared radiation emitted by..."
- 9. I.80 perpendicular to the flight track?

**C9**

- 10. **I.84** ... to the flight track is *around* 150km.
- 11. **Fig. 1** "into the plane of the paper". A paper plane could be an aeroplane made out of paper, which is also nice, given the theme of the study.
- 12. Fig. 1 "Top view in bird perspective of the flightpath" » "top-down view of the flight path"
- 13. I.111 I know what is meant here, but perhaps it is clearer to say "between 20 to 60km"
- 14. **I.112** discussed » conducted.
- 15. I.117 used retrieval » retrieval used
- 16. **I.130** Could the authors provide a citation for readers outside of the field who are not familiar with the 4-D var method for data assimilation?
- 17. I.134 close to reality » useful realistic
- 18. **I.139** was » has been
- 19. **I.140** recent » last
- 20. **I.142** "Though the dynamical core would in principal allow to resolve waves ... " » "Though the dynamical core could in principal resolve waves ... "
- 21. I.146 and 147 ECMWF operational analysis fields
- 22. **I.227** "a differential optical absorption spectrometer" it is sad that this instrument did not get an exciting acronym like the others. Perhaps "DEXTAR" the Differential EXperimental opTical Absorption spectrometeR?
- 23. I.345 shifts a bit » shift

- 24. I.348-349 "These difference help to explain why a joint retrieval ..."
- 25. **I.420** barely » just
- 26. **Fig. 9** If the authors happen to regenerate this figure, perhaps the dots could be plotted on top of the black lines, they are quite hard to see.
- 27. Fig. 9 " ... indicate which wave properties the waves, which do not reach the surface, have at flight altitude." » "... indicate the wave properties at flight altitude for waves that do not reach the surface."
- 28. **Fig. 10** "exemplary" means an exceptionally good example. Perhaps "example" would be better here?
- 29. **I.392** "To investigate, how accurate the forward ray-tracing calculations of the GROGRAT model are, ..." » "To investigate the accuracy of the forward ray-tracing calculations of the GROGRAT model, ..."

---

## Referee Comment (RC2) · Anonymous Referee #2 · 1 Jul 2020

The manuscript describes the investigation of gravity waves in measurements from an airborne campaign over Scandinavia. This constitutes an interesting case study, with original measurement techniques providing an exceptionnal three-dimensional description of mesoscale gravity waves in the lowermost stratosphere. The study serves two purposes: 1. it illustrates a complex wave situation, which includes wavepackets on several scales (a main wave packet with wavelength 400 km, but also a complex background of shorter waves). 2. It demonstrates the capability of the GLORIA instrument to document a complex, multi-scale set of gravity waves. The manuscript is well written, the figures are well prepared, the references are relevant. Publication is recommended once the minor points below are addressed.

[Figure]

Minor Points

l38 the technique can only retrieve wave characteristics in the UTLS? Is there something specific that imposes that the technique can only work in that altitude range?

l46: should the method of Schoon and Zuelicke (2018) also be mentionned here? Or is it intended mainly for model output and could not contribute with observations? For completeness, it could be mentionned in any case.

Caption of Fig 1: I do not fully understand what is meant by 'cover the dark grey area with the LOS.'? (Bythe way, LOS is introduced without explanationn in this first sentence, it is spelled out with the acronym in parentheses a few lines later)

Caption of Fig 1.1: the sentence explaining the parabolic shape of the different lines of sight ("The line-of-sight (LOS), which is a straight line in reality, has a parabolic shape in this plot due to the transformation into a Cartesian coordinate system with the x-axis following the Earth surface.") should come earlier in the caption. It is the basis of the figure. Once that is explained, the meaning of the colored dots, of dark and grey shade areas can be explained, they become much clearer.

l237: 'the complete west coast' -> 'the west coast'?

l245: in km/h ($\text{km}\,\text{h}^{-1}$) there should be a space between km and h.

Figure 5: a color different than the light yellow should be chosen for the second ECMWF curve. It is too hard to see.

l350: do the authors judge that such an extension of the S2D

l426 '=limited angle tomography' : add a parentheses recalling the acronym (LAT) as it was used earlier in the paper. This may help the reader who is starting with the conclusion and then working back through the rest of the paper.

How confident are the authors in the checkerboard pattern that remains after removal of the large-scale wave signal?

l440: the conclusion ("Therefore") that the large-scale wave packet results of several sources is not the appropriate statement supported by the investigation: rather, the investigation and ray-tracing carried out does not allow to decide on the source of the waves. It is possible, or even plausible, that several sources contributed to the observed waves.

l449: again, perhaps recall the acronym (LAT); note that a hyphen is used here (limited-angle tomography), but in other instances it is not. The authors should homogenize.

===================================================== Schoon and Zuelicke, 2018): A novel method for the extraction of local gravity wave parameters from gridded three-dimensional data: description, validation, and application, ATMOSPHERIC CHEMISRY AND PHYSICS, 18, 9, 6971-6983 DOI: 10.5194/acp-18-6971-2018

---

## Author Comment (AC1) · 6 Aug 2020

Dear Editor, Dear Reviewers,

Thank you very much for your effort and these very helpful comments that significantly helped to improve the manuscript. Below you can find a point-by-point reply to all reviewer comments.

Sincerely,

Isabell Krisch

**1 Point-by-point reply to Reviewer #1**

**Referee comment:** Abstract, I.12-13: The phrasing is a bit strange here. If the AIRS measurements barely resolve the wave, how can they support the results? The word "barely" is usually used in a negative context to dismiss something rather than use it to support something else, so I'd suggest rephrasing to: "Co-located AIRS measurements in the middle stratosphere are also in agreement with these results, despite their coarser vertical resolution compared to GLORIA measurements."

Authors' response: Sentence has been rephrased as proposed.

**Referee comment:** I.28: If relevant, I think the authors might like to include Vosper and Ross (2020) here, which has some important considerations for gravity wave momentum flux measurements from near-vertical profiles.

Authors' response: The text has been edited including Vosper and Ross (2020).

**Referee comment:** I.47: The 3-D S-transform method is fully described, tested and validated in Hindley et al. (2019) so this would be a useful reference to included here. This is the planned 'method paper' than underlines the 3DST approach we applied in ?, it just took me a little longer to finish it!

Authors' response: A reference to Hindley et al. (2019) has been added.

**Referee comment:** Introduction The introduction could benefit from a brief synopsis of the paper at the end, since the next section is quite technical. In particular, it would be nice to describe the aircraft campaign a little before hitting the reader with a wall of technical detail. Perhaps something like "Here we analyse airborne GLORIA measurements from a flight over Scandinavia in 2016. We apply ray-tracing techniques to our

**ACPD**
measurements and compare our results to satellite observations and reanalysis. The datasets, spectral analysis and ray-tracing methods are described in Sect. 2. The flight campaign and synoptic conditions are described in Sect. 3... etc... "

**Authors' response:** An additional paragraph outlining the content of each section has been added to the introduction.

**Referee comment:** Fig. 1 The figure is nice but the caption is quite confusing. I would perhaps suggest to start with: "Vertical (a) and horizontal (b) cross-sections of the limb sounding geometry of airborne GLORIA measurements using the LAT configuration. Measurements are made at the tangent points shown by the coloured dots, where the colour indicates the tangent point altitude. In panel (a), ... "

Authors' response: The caption of Figure 1 has been rephrased to enhance readability.

**Referee comment:** Fig. 1 "Images taken under 90âU e azimuth cover the dark grey area with the LOS" - this sentence does not make sense, please rephrase.

Authors' response: The caption of Figure 1 has been rephrased to enhance readability.

**Referee comment:** Fig. 2 The authors should make it clearer how this AIRS sensitivity function is derived, in particular how the effect on measured vertical wavelength is found. Was this function derived mathematically, or were synthetic waves created, then passed through the AIRS vertical retrieval, then measured with the 1-D fitting routine they describe to compare vertical wavelengths in/out?

Is this the same approach as the correction factor applied in the supporting information of Ern et al. (2017)?

**ACPD**
Either way, I think it is important to briefly state how this is derived. It's not made clear in the text, or in the references given, how the stretching of the vertical wavelength takes place mathematically (apologies if I have missed it in the references to Meyer et al. (2018) and Ern et al. (2017)). Also, is this stretching the same for all altitudes in the AIRS retrieval? I am confident that the authors are correct, but it needs to be made clearer in order to convince the impartial reader. Only one or two sentences are required.

**Authors' response:** A description has been added to the manuscript. Synthetic waves were created and their temperature profiles were convolved with an exemplary AIRS averaging kernel. Wavelength and amplitude of the perturbations in the resulting temperature profile were estimated with a 1-D fitting routine and compared to the original values. This convolution with the averaging kernel matrix is a linearization of the real AIRS retrieval and in this a simplification. However, due to the temperature perturbations being small compared to the absolute temperature values the authors think that such a linearization can be applied here. The correction factors applied in the supporting information of Ern et al. (2017) were calculated with the same method but by applying a full AIRS temperature retrieval experiment.

**Referee comment:** I.118-120 Looking again at Fig. 2, I think the threshold should be 25km because above this the waves are slightly shrunk, which would introduce a low bias in GWMF results. AIRS is not the focus of this study, but for information I find that in practice the vast majority of vertical wavelengths measured in AIRS are generally between 15 and 25km (e.g. Hindley et al., 2019). Shorter vertical wavelengths tend to be lost below the noise threshold, and longer vertical wavelengths tend to be underestimated due to only a few cycles being present in the vertical, which reduces the signal to noise. We did not apply a correction in the measured vertical wavelength in Hindley et al. (2019) because I found that the measurement error in the vertical wavelength of around 10-25% was comparable to the correction factor, so I didn't want to
incorrectly apply it and introduce a further source of error. Here, error in the vertical wavelength measurement of the S3D method could be similar, so they may have the same problem, but the authors do not discuss or quantify that here. Anyway, AIRS is not the focus of the paper. Perhaps, to avoid further discussion, the last two sentences could be changed to simply: "For vertical wavelengths below 25km, the temperature amplitude is significantly underestimated and measured vertical wavelengths in AIRS can appear up to 45% larger than their true value. As such, AIRS measurements of vertical wavelengths below 25km may be over-estimated, so caution is advised."

Authors' response: The two sentences have been rephrased as proposed.

**Referee comment:** I.140 I assume the authors are describing operational analysis here, as distinct from reanalysis? Including the term "operational analysis" would make this clearer. Suggest rephrasing the line to "The ECMWF operational analysis for the year 2016 used here uses...", if that is what is meant.

**Authors' response:** The sentence has been rephrased as proposed and the term "ECMWF analysis" has been replaced by "ECMWF operational analysis" throughout the paper.

**Referee comment:** I.203 I think scale height H and buoyancy frequency N still need to be defined.

**Authors' response:** One sentence has been added to introduce all variables in Equation (3).

**Referee comment:** I.215 It's interesting that the authors used the ECMWF operational analysis for the GROGRAT ray-tracing and not the ERA5 reanalysis. It's a bit strange to interpolate the 6-hour operational analysis in time when ERA5 reanalysis is available
hourly, which has improved accuracy anyway due to more data assimilation. I don't think it affects the results, but it is a limitation. Was it needed to be run in real time during the campaign? If I have missed something, I apologise.

**Authors' response:** During and right after the campaign only ECMWF operational analysis fields were available and thus used for ray-tracing investigations originally. Once ERA5 became available for the investigation period, it was discussed to use these fields instead. However, we think that due to the temporal interpolation within GROGRAT, 6-hourly fields are sufficient to properly construct the atmospheric back-ground. Thus, for the present paper we decided to stay with the common approached of using ECMWF operational analysis fields.

**Referee comment:** I.240 It would be good to include the equation used here to find this number, if possible.

Authors' response: Equation has been added to the manuscript.

**Referee comment:** Fig. 3 The coastline is very difficult to see in grey. Perhaps either apply the same coastline method as in Fig. 12, or maybe the coastline in black and the wind vectors in a bright blue or something? If this is a lot of work, do not worry, the figure is acceptable as it is.

Authors' response: Coastline of Fig. 3 has been adjusted.

**Referee comment:** I.341 Are eastward waves permitted here in the background wind conditions. Given that they must be reasonably slowly moving in order to be measured by GLORIA?

**Authors' response:** In general, GWs which move with the wind are possible, but their ground-based phase velocity would be very high. Thus, they would not be observable

**ACPD**
by GLORIA. The manuscript has been edited for clarification.

**Referee comment:** I.352 Perhaps change "This is beyond the scope of this paper" to "These results provide guidelines for the future development of the S3D method." - Turn a negative into a positive :) Out of interest, how much has the S3D code changed between Lehmann et al. (2012) to Ern et al. (2017) to Krisch et al. (2017) to now?

Authors' response: The sentence has been changes as proposed.

Changes/adaptions made to the S3D code since the original publication of Lehmann et al. (2012) are:

- Implementation of the oblique AIRS geometry for off-nadir soundings; for Ern et al. (2017)
- Recoding of S3D routine in Python (originally IDL) with tests showing equal results for regular longitude × latitude × altitude grids of global model data sets; for Krisch et al. (2017)
- Implementation of X x Y x altitude rectangular grids for GLORIA and local models; for Krisch et al. (2017)
- Usage of units (PINT) within the code; 2019
- Some refactoring of the code for better readability and easier implementation of new features; 2020
- fixed cube size in km for lon-lat grids; 2020
- No changes were made to the S3D fitting algorithm itself (Basin-hopping planned, see below)

Changes planned for the near future:
- release of a Basin-Hopping method as an alternative to steepest gradient and interval nesting, better refitting ( $\chi^2$  > threshold) order
- direct application to ICON grid without re-interpolation (as used currently in Stephan et al 2019)

**Referee comment:** Fig. 12 The largest wave amplitudes measured in the S3D cubes seems to be shifted to the north-east of where the largest wave perturbations are found in Fig. 11b, between +/- 5 degrees longitude. Is there a reason for this or just where the cubes lie?

**Authors' response:** A bug in the plotting script caused this north-east shift. The bug has been corrected.

**Referee comment:** Fig. 12 The authors should briefly discuss the apparent reduction in the S3D measured amplitude in Fig. 12a compared to the temperature perturbation amplitudes in Fig. 11b to the east of Scotland. I am aware that the S3D method underestimates wave amplitudes due to averaging over the cube size but it should be mentioned here too if that is what is going on. Sorry if I have missed it.

**Authors' response:** The colour scale of Fig. 12a has been updated to cover the full range. An explanation on the amplitude underestimation due to the S3D method has been added to the text.

**Referee comment:** I.400-401 Again, the use of "barely" here is not very useful. The panels in Fig. 11 show measurements that are around 9km apart in the vertical. This is slightly more than one atmospheric scale height. Due to the exponential increase in wave amplitude with height due to decreasing density, one might expect that a gravity wave's amplitude would increase by perhaps a factor of 2 or more over this height range. If the colour limits on Fig. 11a were set to +/- 3.5K, rather than around +/- 7K
as in Fig. 11b, then I think many more temperature perturbations would be "visible". The problem is, however, that the desired gravity wave temperature perturbations are too small to be visible above the retrieval noise, as the authors discussed above, due to amplitude attenuation for shorter vertical wavelengths. The authors should perhaps rephrase the sentence to reflect this. If it is straightforward to do, the colour limits of Fig. 11a could also be changed and given its own colour bar to make it more useful.

**Authors' response:** The text has been edited for clarity. The colour scale of Fig. 11a has been changed as proposed.

**Referee comment:** 1.408 Could the authors provide a reference to the exact averaging kernel that was applied here? Also, I don't think a simple multiplication operation is sufficient. I think either the full AIRS vertical retrieval should be run on the ERA5 fields or. at the very least, each height level should be separately convolved with the appropriate averaging kernel for that height and then combined. One cannot apply a resolution kernel just by multiplication alone, I believe convolution is needed (unless that multiplication is in the Fourier domain of course, in which case it is convolution). Sorry if I have misunderstood this from what the authors have written. I do have confidence that the authors have used an appropriate method, but they should describe it more clearly. Neither the exact kernel nor the method is shown anywhere in the paper, or in the references listed (sorry if I have missed it), nor how the stretching of the vertical wavelength that occurs in AIRS measurements arises from the maths. Do they mean the kernels shown in Hoffmann and Alexander (2009)? If so, how are they combined? By the way, I am aware of the current condition of Lars Hoffmann, so if this step was performed by him and the authors cannot ask him then do not worry, a reference the relevant kernels in Hoffmann and Alexander (2009) will be sufficient.

**Authors' response:** An example of the averaging kernels (temperature sensitivity functions) of the used AIRS retrieval has been added to the manuscript (new Fig. 2a). Further, it has been clarified in the text that a convolution of the ERA5 field with these
AIRS averaging kernels was performed here.

**Referee comment:** Fig. 13 caption See above point about multiplication.

Authors' response: Wording has been changed to convolution.

**Referee comment:** I.413 As mentioned above, it would be good to have more information on how the sensitivity function is calculated.

**Authors' response:** A detailed description and an additional plot with the averaging kernel matrix has been added to Section 2.2.

Referee comment: 1.420-421 "GLORIA and AIRS cover rather different parts of the full gravity wave spectrum". If the authors make a revision of the text in Sect. 4.3, I think this should be more of the main theme running through the discussion of results. It could be clearer that, as I understand it, the story is that the authors measure the waves in GLORIA, show that they propagate upwards via forward ray-tracing and are resolved in AIRS data above. Despite the reduced vertical resolution of AIRS, and the inherent amplitude/wavelength errors discussed, the wave is still resolved at 36km, showing that their GLORIA measurements and ray-tracing results are valid. This is a success. The ERA5-as-AIRS data also shows this, which is a further success. Instead, the language used in the AIRS comparison is somewhat defensive and it does not need to be. The two instruments measure different parts of the GW spectrum, they both have their own limitations as with all observations, and here they measure at two different altitudes. As I mentioned above regarding the abstract, I believe the key result here is that the waves are resolved above GLORIA almost exactly as expected despite the resolution difficulties of the AIRS measurements. Again, this presents the results in a positive light rather than being lost in technicalities. Anyway, to me this just shows that the authors have a good attention to detail, so I leave it up to them if they want to adjust ACPD
the text.

**Authors' response:** The paragraph including the AIRS results in Sect. 4.3 has been rephrased along the proposed suggestion.

**Referee comment:** I.443-446 See comment above. Here, I would suggest that the order the two sentences is reversed. This presents the results positively: "Gravity wave signals in AIRS measurements agree qualitatively very well with the predictions by the ray tracer and ERA5. This is despite a strong underestimation of wave amplitudes for waves with vertical wavelengths shorter than around 25km. Furthermore, we report an overestimation of the vertical wavelengths for AIRS measurements. "However, I would suggest not including the last sentence I wrote above unless the authors have shown (or more clearly referenced) how the vertical wavelength underestimation occurs in AIRS measurements.

**Authors' response:** The paragraph including the AIRS results in Sect. 4.3 has been rephrased along the proposed suggestion in this and the previous comment.

**Referee comment:** Appendix A I just want to mention that the inclusion of Appendix A is very good, particularly the frequency response figures for different filters for GW analysis. Many studies simply choose a method and plot the results, so it is very important that filtering choices here are properly assessed, as the authors have done here.

Authors' response: Thanks.

**Referee comment:** I.510 Is there a reason the authors chose the SG-filter over the Butterworth? The SG-filter has quite a bit of unwanted "ringing" in the pass-band, which the Butterworth does not. However I can see in Fig. A2 that the Butterworth
does some strange "shifting" of the 2-D perturbations which the SG-filter does not.

**Authors' response:** Fig. A2 shows slightly better results for the SG-filter (no strange shifting) and also the Pearson coefficients are better for SG for all wave packets which were tested (not only the ones shown in the Appendix). Based on this the decision was taken to use SG.

**2 Point-by-point reply to Reviewer #2**

**Referee comment:** I38 the technique can only retrieve wave characteristics in the UTLS? Is there something specific that imposes that the technique can only work in that altitude range?

**Authors' response:** The method in general also works at higher altitudes. However, the volume retrievable by limb sounding is always limited to below the instrument. As the instrument carrier is an aircraft flying in the lowermost stratosphere only the UTLS region can be investigated.

The manuscript has been updated to specify that GLORIA is an airborne instrument.

**Referee comment:** 146: should the method of Schoon and Zülicke (2018) also be mentioned here? Or is it intended mainly for model output and could not contribute with observations? For completeness, it could be mentioned in any case.

**Authors' response:** In choosing a spectral analysis method you have to find the best compromise between spectral resolution and spatial resolution for your data and your analysis purpose (cf. Heisenberg's uncertainty relation). Fourier transform, on the one end of the scale, provides full spectral resolution but no spatial resolution. Hilbert transform (as used by Schoon and Zülicke, 2018) provides perfect spatial resolution

ACPD
but has to rely on the assumption that at a given location only a single wave and no further disturbances exist. Methods in between these two extrema are windowed FFT, Wavelet, S-transform and S3D (from former to latter with increasing spatial and decreasing spectral resolution).

The UWADI tool from Schoon and Zülicke (2018) is developed and tested for horizontal divergence of winds. This cannot be applied to GLORIA temperature observations. In principle, however, a Hilbert transform could be applied to temperature observations and GWMF inferred from the wave vector as well. However, for temperature data the stability towards noise and incomplete background removal would have to be tested and also whether the single wave assumption induces uncertainties in the wave vector and hence in GWMF.

We believe that S3D is, for our purpose, the best compromise between spectral and spatial resolution and has been proven to be reasonably robust on the superposition of a few waves and noise.

For completeness a short summary of the above has been added to the manuscript.

**Referee comment:** Caption of Fig 1: I do not fully understand what is meant by 'cover the dark grey area with the LOS.'? (By the way, LOS is introduced without explanation in this first sentence, it is spelled out with the acronym in parentheses a few lines later)

**Authors' response:** The caption of Figure 1 has been rephrased to enhance readability.

**Referee comment:** Caption of Fig 1.1: the sentence explaining the parabolic shape of the different lines of sight ("The line-of-sight (LOS), which is a straight line in reality, has a parabolic shape in this plot due to the transformation into a Cartesian coordinate system with the x-axis following the Earth surface.") should come earlier in the caption.
It is the basis of the figure. Once that is explained, the meaning of the colored dots, of dark and grey shade areas can be explained, they become much clearer.

Authors' response: The caption of Figure 1 has been rephrased to enhance readability.

**Referee comment:** l237: 'the complete west coast' - > 'the west coast'? **Authors' response:** Sentence has been rephrased as proposed.

Referee comment: I245: in km/h there should be a space between km and h.

Authors' response: All units have been revisited and updated were necessary.

**Referee comment:** Figure 5: a color different than the light yellow should be chosen for the second ECMWF curve. It is too hard to see.

Authors' response: Figure 5 has been adjusted accordingly.

Referee comment: I350: do the authors judge that such an extension of the S2D

**Authors' response:** Detailed simulation studies would be required to be able to judge if and how an extension of the S3D routine towards checkerboard patterns can provide reasonable results. Such studies are planned for the near future, but are beyond the scope of our current analysis.

**Referee comment:** I426 '=limited angle tomography' : add a parentheses recalling the acronym (LAT) as it was used earlier in the paper. This may help the reader who is starting with the conclusion and then working back through the rest of the paper.

ACPD
Authors' response: Abbreviation has been added.

**Referee comment:** How confident are the authors in the checkerboard pattern that remains after removal of the large-scale wave signal?

**Authors' response:** A visual inspection of fitted wave pattern (first wave component) looks very reasonable. The checkerboard pattern (difference between retrieval and first wave component) also looks physical and not like noise or fitting or measurement artefacts. Thus, the authors are confident that the shown checkerboard pattern depicts reality.

**Referee comment:** I440: the conclusion ("Therefore") that the large-scale wave packet results of several sources is not the appropriate statement supported by the investigation: rather, the investigation and ray-tracing carried out does not allow to decide on the source of the waves. It is possible, or even plausible, that several sources contributed to the observed waves.

Authors' response: The text has been edited for clarity.

**Referee comment:** I449: again, perhaps recall the acronym (LAT); note that a hyphen is used here (limited angle tomography), but in other instances it is not. The authors should homogenize

**Authors' response:** The term limited angle tomography has been homogenized throughout the text and abbreviations have been added were missing.
**References**

- Ern, M., Hoffmann, L., and Preusse, P.: Directional gravity wave momentum fluxes in the stratosphere derived from high-resolution AIRS temperature data, Geophys. Res. Lett., 44, 475– 485, https://doi.org/10.1002/2016GL072007, 2017.
- Hindley, N. P., Wright, C. J., Smith, N. D., Hoffmann, L., Holt, L. A., Alexander, M. J., Moffat-Griffin, T., and Mitchell, N. J.: Gravity waves in the winter stratosphere over the Southern Ocean: high-resolution satellite observations and 3-D spectral analysis, Atmospheric Chemistry and Physics, 19, 15377–15414, https://doi.org/10.5194/acp-19-15377-2019, 2019.
- Hoffmann, L. and Alexander, M. J.: Retrieval of stratospheric temperatures from Atmospheric Infrared Sounder radiance measurements for gravity wave studies, J. Geophys. Res., 114, D07 105, https://doi.org/10.1029/2008JD011241, 2009.
- Krisch, I., Preusse, P., Ungermann, J., Dörnbrack, A., Eckermann, S. D., Ern, M., Friedl-Vallon, F., Kaufmann, M., Oelhaf, H., Rapp, M., Strube, C., and Riese, M.: First tomographic observations of gravity waves by the infrared limb imager GLORIA, Atmos. Chem. Phys., 17, 14 937–14 953, https://doi.org/10.5194/acp-17-14937-2017, 2017.
- Lehmann, C. I., Kim, Y.-H., Preusse, P., Chun, H.-Y., Ern, M., and Kim, S.-Y.: Consistency between Fourier transform and small-volume few-wave decomposition for spectral and spatial variability of gravity waves above a typhoon, Atmos. Meas. Tech., 5, 1637–1651, https://doi.org/10.5194/amt-5-1637-2012, 2012.
- Meyer, C. I., Ern, M., Hoffmann, L., Trinh, Q. T., and Alexander, M. J.: Intercomparison of AIRS and HIRDLS stratospheric gravity wave observations, Atmos. Meas. Tech., 11, 215–232, https://doi.org/10.5194/amt-11-215-2018, 2018.
- Schoon, L. and Zülicke, C.: A novel method for the extraction of local gravity wave parameters from gridded three-dimensional data: description, validation, and application, Atmospheric Chemistry and Physics, 18, 6971–6983, https://doi.org/10.5194/acp-18-6971-2018, https://www.atmos-chem-phys.net/18/6971/2018/, 2018.
- Vosper, S. and Ross, A.: Sampling Errors in Observed Gravity Wave Momentum Fluxes from Vertical and Tilted Profiles, Atmosphere, 11, https://doi.org/10.3390/atmos11010057, 2020.
- Wright, C. J., Hindley, N. P., Hoffmann, L., Alexander, M. J., and Mitchell, N. J.: Exploring gravity wave characteristics in 3-D using a novel S-transform technique: AIRS/Aqua measurements over the Southern Andes and Drake Passage, Atmos. Chem. Phys., 17, 8553–8575, https://doi.org/10.5194/acp-17-8553-2017, 2017.